# Hybrid architectures for terahertz molecular polaritonics

Ahmed Jaber [1,2], Michael Reitz [2,3,4], Avinash Singh[1,2], Ali Maleki[1,2], Yongbao Xin[5], Brian T. Sullivan[5], Ksenia Dolgaleva [1,2,6], Robert W. Boyd[1,2,6,7], Claudiu Genes[2,3,4] ✉ & Jean-Michel Ménard [1,2,6] ✉

Atoms and their different arrangements into molecules are nature's building blocks. In a regime of strong coupling, matter hybridizes with light to modify physical and chemical properties, hence creating new building blocks that can be used for avant-garde technologies. However, this regime relies on the strong confinement of the optical field, which is technically challenging to achieve, especially at terahertz frequencies in the far-infrared region. Here we demonstrate several schemes of electromagnetic field confinement aimed at facilitating the collective coupling of a localized terahertz photonic mode to molecular vibrations. We observe an enhanced vacuum Rabi splitting of 200 GHz from a hybrid cavity architecture consisting of a plasmonic meta-surface, coupled to glucose, and interfaced with a planar mirror. This enhanced light-matter interaction is found to emerge from the modified intracavity field of the cavity, leading to an enhanced zero-point electric field amplitude. Our study provides key insight into the design of polaritonic platforms with organic molecules to harvest the unique properties of hybrid light-matter states.

Strong light–matter interactions can modify the fundamental properties of some physical systems, leading to applications beyond fundamental science. In general, the light modes used in such experiments range from the visible to the region of mid-infrared. In the visible regime, where light couples to electronic transitions, experiments have shown modifications of charge conductivity[1,2], photochemistry[3,4], and single-molecule branching ratios[5,6]. The condition for observing coherent exchanges between light and matter requires the coupling strength, also known as the vacuum Rabi splitting (VRS), to exceed all of the various loss rates in the system. To mitigate optical loss, high finesse optical resonators are utilized. Standard designs in early pioneering efforts have made use of distributed Bragg reflector (DBR) microcavities with active media consisting of either an inorganic semiconductor quantum well design[7] or organic semiconductors[8,9].

Within regions of lower energies within the electromagnetic spectrum, light can also be directly coupled to intra- and inter-molecular vibrational modes[10–12]. This can lead to emerging applications in chemistry through the modification of ground state potential landscapes, allowing for the manipulation of chemical reaction pathways and rates[13–15]. However, for the purpose of studying light–matter coupling involving far-infrared light and low-energy vibrational transitions of molecules, the DBR design becomes unpractical since most dielectrics are absorptive in this region, and their required thickness exceeds the capacity of most nanofabrication equipment. In the THz region (wavelengths from 0.1 to 1 mm), the coupling to low energy bonds, such as intermolecular vibrations or hydrogen bonding, becomes accessible[16,17]. At these larger wavelengths, Fabry–Perot (FP) cavities utilizing metallic planar mirrors have been demonstrated,

[1]Department of Physics, University of Ottawa, Ottawa, ON K1N 6N5, Canada. [2]Max Planck Centre for Extreme and Quantum Photonics, Ottawa, ON K1N 6N5, Canada. [3]Max Planck Institute for the Science of Light, D-91058 Erlangen, Germany. [4]Department of Physics, Friedrich-Alexander-Universität Erlangen-Nürnberg, D-91058 Erlangen, Germany. [5]Iridian Spectral Technologies Ltd., Ottawa, ON K1G 6R8, Canada. [6]School of Electrical Engineering and Computer Science, University of Ottawa, Ottawa, ON K1N 6N5, Canada. [7]University of Rochester, Rochester, NY 14627, USA. ✉e-mail: claudiu.genes@mpl.mpg.de; jean-michel.menard@uottawa.ca

where a VRS on the order of 68 GHz has been reached[18]. Following this framework, more sophisticated cavity architectures can potentially lead to even stronger polaritonic splitting with molecular systems in the THz region.

In the context of light–matter interaction with solid-state emitters, different types of THz resonators based on plasmonic emitters/antennas[19,20], waveguides[21,22], photonic crystal[23], and metasurfaces (MSs)[24–26], or a combination of them[27–30], have been explored to achieve strong field confinement. However, THz cavity architectures involving multiple electromagnetic resonators have not yet been explored in combination with resonant molecular ensembles, although few of these structures optimized for the mid-infrared range have been investigated[31,32]. Thus, a systematic study of metasurface-based cavities in the emerging context of THz molecular polaritonics is needed to enable a range of chemistry applications such as those discussed above.

Here, we propose a set of four THz resonator architectures illustrated in Fig. 1, including two hybrid designs relying on a reflective MS (analogous, e.g., to subradiant optical mirrors realized with trapped atoms[33]) replacing the planar mirror of a FP cavity to achieve better cavity finesse[34]. These designs are used to yield unprecedented electromagnetic field spatial distributions peaking at the interface of the MS. We explore the coupling of these fields to a resonant transition of glucose, a molecule with strong relevance to biological processes, including metabolism and photosynthesis. Until now, the coupling of molecules with hybrid cavities has remained mostly unexplored, in part due to the complexity of fabricating a dense layer of organic material within a spatially confined electromagnetic mode volume. Using a spray coating technique[35], we obtain a dense molecular layer in direct contact with the surface of the MS, allowing us to reach the strong light–matter coupling regime involving a resonant plasmonic and molecular vibrational mode. This system can then be inserted into the design of a planar resonator geometry to enable further hybridization with a photonic mode. We experimentally and numerically compare the transmission properties of such cavity architectures, exploring the evolution from a standalone resonator towards a hybrid

design with and without (see Supplementary Information (SI), section Empty hybrid cavity) the presence of a resonant molecular transition. This systematic experimental study, supported by theory and simulations, provides a unique window into the complex interaction between plasmonic, photonic, and vibrational resonances. Most notably, we compare the polaritonic splitting achieved with a plasmonic MS and a standard FP cavity design. Interestingly, a similar Rabi splitting is observed, although fewer molecular emitters can be contained within the evanescent mode volume of the MS, in comparison to those within the FP cavity mode. This indicates that the metasurface, through a larger mode confinement and stronger field enhancement, enables an enhanced coupling per molecule. Then, using a hybrid cavity architecture, where the MS replaces one of the mirrors of the FP cavity, we reach a 40% larger polaritonic splitting with less glucose volume than a standard FP cavity due to a modified intracavity field.

## Results and discussion
### Setup and approach
MSs are constructed using a photolithography and metal evaporation process in which arrays of cross-shaped aluminum elements are fabricated onto a THz-transparent substrate. The dimensions of the metallic elements and the periodicity of the array, schematically shown in Fig. 1a, are optimized to achieve a sharp THz plasmonic resonance, causing a distinctive dip in the spectral transmission. The four-fold symmetry of the cross-shaped elements is chosen to ensure a response insensitive to the polarization state of the THz radiation. In the first experiment, the layer is deposited directly on the MS as shown schematically in Fig. 1a. In a second experiment, two partially reflective mirrors ($R \approx 0.85$), fabricated by sputtering 9 nm of gold on a semiconductor substrate, form a FP cavity in the THz region (Fig. 1b). In a third and fourth experiment, we explore hybrid architectures, schematically shown in Fig. 1c, d, where one mirror of the FP cavity is replaced by a MS. We investigate the THz transmission response when a glucose layer is deposited on the planar mirror (H1) or on the MS (H2).

Characterization of the different architectures is performed with a THz time-domain spectroscopy (THz-TDS) technique, depicted in the setup shown in Fig. 2. For our investigation of the vibrational resonances of glucose interacting with THz resonators, we focus our measurement range on the low THz frequency range between 0.5 and 3 THz. In this range, we find that EOS is a more suitable detection technique than the standard Fourier-transform infrared (FTIR) spectroscopy, which is broadly used for sample characterization in the mid-IR. We found that EOS offers a larger signal-to-noise ratio than THz-FTIR systems we tested around the bare resonance frequency of 1.43 THz in our experiments.

In order to observe a sizeable VRS, both the linewidths of the photonic device and that of the molecular ensemble have to be optimized. Saccharides, like glucose, are known to have a distinctive narrow radiative transition in the THz region originating from collective intermolecular vibrations due to the hydrogen bonding networks in the crystalline phase[36]. The complex refractive index of glucose is measured with time-resolved THz spectroscopy. The inset of Fig. 2 shows the absorption spectrum of glucose with a prominent vibrational resonance at 1.43 THz and background absorption increasing towards higher frequencies. We utilize phase and amplitude information provided by the EOS detection technique to directly extract the real and imaginary parts of the refractive index of glucose (see the "Methods" section). The real part of the refractive index of glucose is $n_{gl} = 1.9$ at 1.43 THz. We used these values to model the spectral transmission of the MS with the Lumerical FDTD solver (Lumerical Inc.) and the response of the devices, including a photonic cavity with a theoretical analysis based on the transfer matrix method (see SI, section Transfer matrix theory). The description of the interaction between an electromagnetic mode (angular frequency $\omega_c$) and an

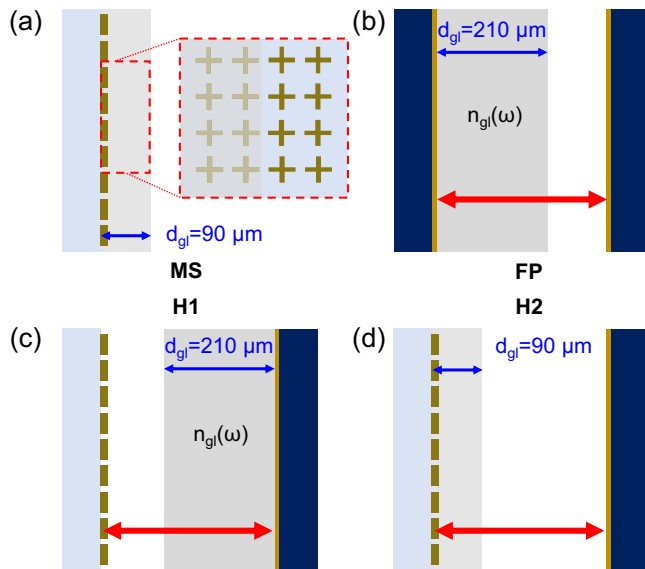

**Fig. 1 | Hybrid architectures for light–matter coupling. a** Plasmonic metasurface (MS) as an array of cross-shaped metal elements coated with a $(90 \pm 10)$ μm-thick glucose layer. **b** Fabry–Perot (FP) cavity where one mirror is coated with a $(210 \pm 10)$ μm-thick glucose layer. **c** and **d** Hybrid cavity designs in which one mirror of the FP cavity is replaced by a MS, and where the glucose layer either covers the mirror (H1) or the MS (H2). The glucose thickness is labeled as $d_{gl}$ and the corresponding refractive index is labeled as $n_{gl}(\omega)$.

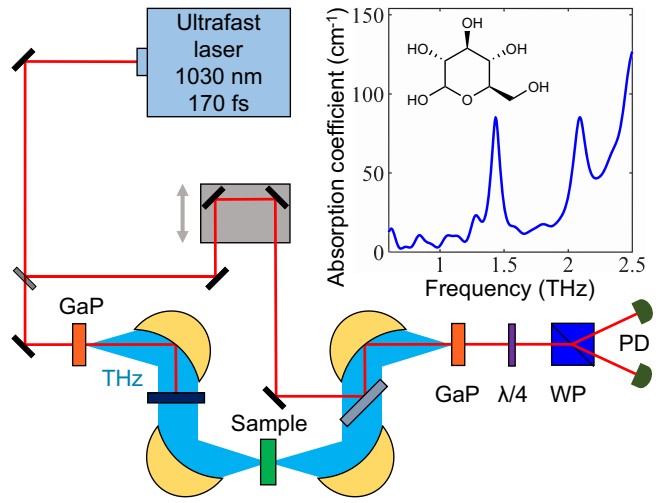

**Fig. 2 | THz-TDS setup and the THz absorption coefficient of glucose.** Schematic of the THz time-resolved spectroscopy setup. An ultrafast laser source is used to generate THz through optical rectification in a GaP crystal. The detection process uses a partially reflected pulse from the same optical source to perform standard electro-optic sampling (EOS) inside another GaP crystal. In brief, the THz electric field is revealed by resolving THz-induced birefringence in the near-infrared gating pulse, which is monitored as a function of time delay with the THz pulse with a quarter-wave plate ($\lambda/4$), Wollaston prism (WP) and a pair of balanced photodiodes (PD). (Inset) Absorption spectrum of a 300 μm-thick glucose ($C_6H_{12}O_6$) pellet measured with time-resolved THz spectroscopy and featuring a prominent vibrational resonance at 1.43 THz.

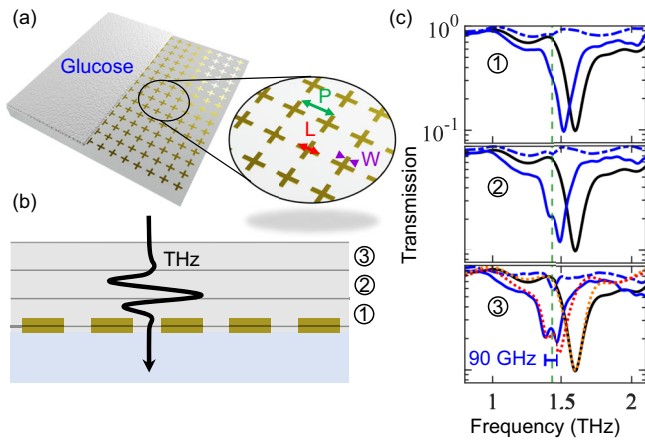

**Fig. 3 | Strong light–matter coupling with glucose-coated MS. a** Schematic of an MS designed from an array of cross-shaped aluminum elements (shown in dark yellow color for clarity) with a glucose coating covering half the structure. The inset is a zoom-in defining the structural dimensions: the periodicity ($P$), cross-arm length ($L$), and cross-arm width ($W$), which are optimized to provide a narrow plasmonic resonance. **b** A cross-sectional schematic of the MS with three thicknesses of glucose layers: (1) 30 μm, (2) 60 μm, and (3) 90 μm, deposited with successive spray coating passes. **c** THz-TDS measurements taken of these three structures (blue line) and the glucose layer on the bare substrate (without the MS) (blue dashed line). The transmission spectrum of the uncoated MS is provided for comparison (black line). The vertical green dashed line shows the vibrational resonance frequency of the targeted glucose mode. In (3), we overlap transfer matrix calculations over the coupled metasurface and bare metasurface experimental measurements (red and orange dotted lines, respectively).

ensemble of $N = \rho V$ (in volume $V$ with number density $\rho$) molecular vibrational dipoles (angular vibrational frequency $\nu$) is given by the Tavis–Cummings Hamiltonian (setting $\hbar = 1$)

$$H_{TC} = \omega_c \hat{a}^\dagger \hat{a} + \sum_j \nu \hat{b}_j^\dagger \hat{b}_j + \sum_j g_j (\hat{a}^\dagger \hat{b}_j + \hat{b}_j^\dagger \hat{a}), \tag{1}$$

where $\hat{a}$, $\hat{b}_j$ are the bosonic annihilation operators for the cavity mode and the $j$th molecular dipole, respectively. The coupling $g_j = \mu \mathscr{E} f(r_j)$ $\boldsymbol{\varepsilon}_\mu^j \cdot \boldsymbol{\varepsilon}_c$ is given by the product of the dipole moment of the vibrational transition $\mu$, the zero-point electric field amplitude $\mathscr{E}$ (inversely proportional to $1/\sqrt{V_{opt}}$—where $V_{opt}$ is the cavity mode volume) and a spatial function $f(r_j)$ evaluated at the position of the molecule $r_j$. The unit vectors $\boldsymbol{\varepsilon}_\mu^j$ and $\boldsymbol{\varepsilon}_c$ account for the relative orientation between the molecular dipole and the cavity polarization, respectively. Glucose molecules do not have a preferential orientation in our experiments since they are deposited by a spray coating technique. Furthermore, the metasurface design utilized as standalone and in the hybrid cavities is created to have 4-fold symmetry, which allows its transmission spectrum to be independent of the incident THz polarization state. As a result, changing the orientation of the incident polarization relative to the metasurface does not have any impact on the results. Assuming, for simplicity, the case where all couplings are identical $g_j = g$, the Hamiltonian describes the coupling of a single bright mode $\hat{B} = \sum_j \hat{b}_j / \sqrt{N}$ to the cavity field with collective coupling strength $g_N = g\sqrt{N}$. This is the so-called vacuum Rabi splitting (to be precise, the splitting between the peaks is $2g_N$) and describes the strength of the light–matter coherent exchanges. In the limit where this rate dominates any loss processes, the system is said to be in the collective strong coupling regime. We remark that the Hamiltonian presented in Eq. (1) assumes the rotating wave approximation (RWA) which is valid if the collective coupling strength is much smaller than the transition frequency.

A very simple alternative to derive the Rabi splitting, even beyond the RWA, is to perform a linear response analysis via the transfer matrix approach. In this way, the splitting is obtained as the difference between the light–matter hybridized states (polaritons) in the transmission profile as a function of the incoming laser frequency (see SI, section "Transfer matrix theory").

## Coupling with a metasurface resonator

The spectral transmission of a plasmonic MS depends not only on the geometry and periodicity of its metallic elements but also on the background media surrounding its interface. The design shown in Fig. 1a presents a THz-transparent substrate underneath the MS and glucose or air on the top surface. The structure is optimized to allow strong coupling between the molecular resonance at 1.43 THz and a resonant plasmonic mode. A rendered depiction of the sample is shown in Fig. 3a with an inset zoom-in that defines the MS geometry. Since the coupling strength scales with the filling fraction (FF) $FF = \sqrt{N/V_{opt}} = \sqrt{\rho V / V_{opt}}$, the glucose layer must ideally fill up the plasmonic mode volume to ensure that the largest number of emitters are coupled with the electromagnetic mode. Further addition of glucose will not increase the coupling, which is obviously limited by the density $\rho$, but instead only brings detrimental effects due to the increased absorption. We, therefore, iteratively deposit thin layers of glucose (height $d_{gl}$, cross-section $A$) while monitoring the transmission properties. A cross-sectional diagram of the coated MS is illustrated in Fig. 3b, where three coating thicknesses are picked to showcase the improvement of the light–matter interaction. We make use of a spray coating technique, which allows us to deposit thin films of crystalline molecules with subwavelength grain structures directly within the plasmonic mode volume of our metasurfaces (see the "Methods" section for procedure outline). The challenge of studying organic molecular polaritonics with a THz MS is not the fabrication of the metasurface but rather in depositing a sufficiently high density of organic molecules at the interface of the MS to allow sufficient coupling strengths; our technique fulfills this condition while being robust.

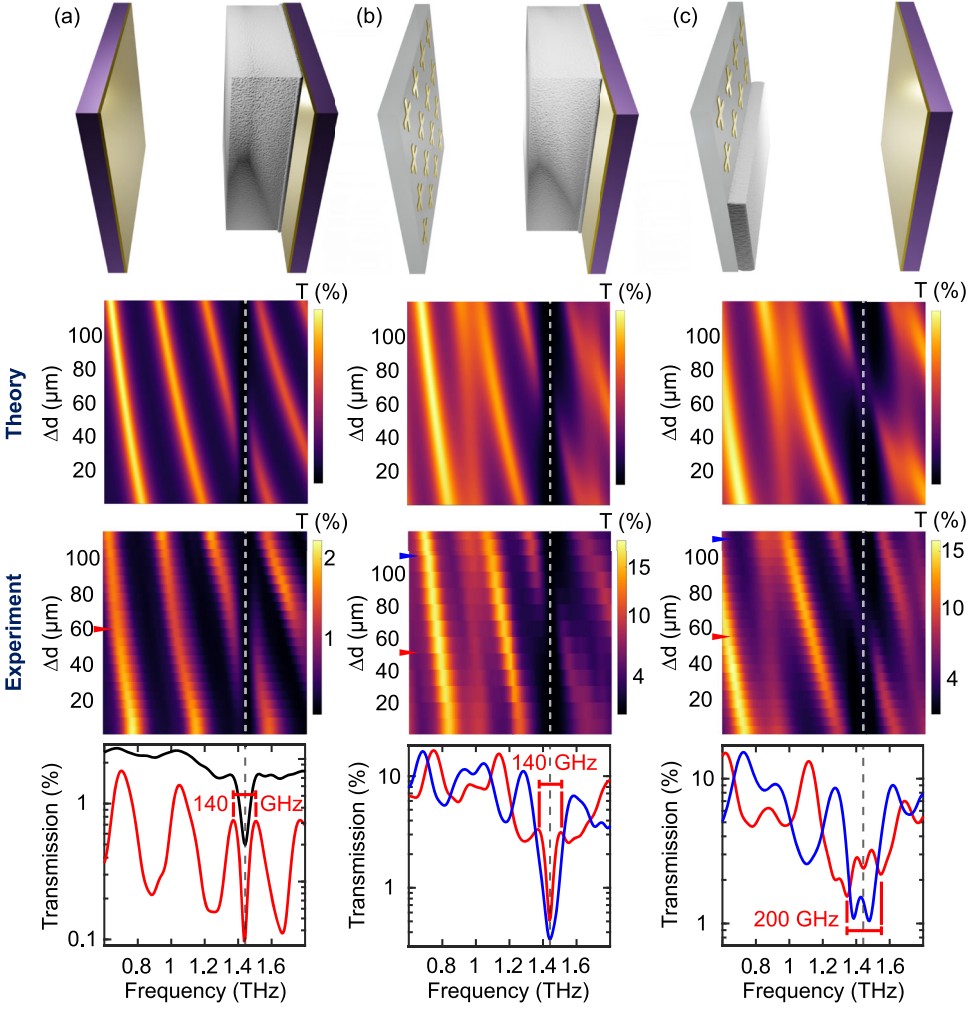

**Fig. 4 | Strong light–matter coupling with FP and hybrid cavity architectures.** Transmission spectra for **a** standard FP cavity, **b** H1 and **c** H2. Top to bottom row shows the schematics of the cavity architectures, the 2D transmission map calculated with the transfer matrix approach as a function of relative cavity spacing and frequency, the experimentally acquired transmission maps of the cavities, and cross sections through the transmission profile of the experimental results. In the plots at the bottom, the black curve in **a** shows the transmission of a glucose-coated gold mirror, while the red and blue curves show the transmission for different cavity lengths (on-resonance/off-resonance), as indicated by the arrows in the transmission map above. The H2 architecture leads to an enhanced polaritonic response, showing a substantially broader Rabi splitting when a cavity mode is overlapping with the polaritons of the coupled MS.

The complex frequency-dependent transmission coefficient for the hybridized MS-glucose system can be obtained from transfer matrix theory as (neglecting the effects of the substrate)

$$t_{MS}(\omega) = \frac{e^{-i\omega d_{gl} n_{gl}(\omega)/c}}{1 - i\zeta_{hyb}(\omega)}, \quad (2)$$

with $c$ the speed of light, $n_{gl}(\omega)$ is the refractive index of glucose, and $\zeta_{hyb}$ describes the so-called polarizability of the hybridized MS. The expression for $\zeta_{hyb}$, derived using a coupled-dipoles model (see SI, section "Coupled-dipoles approach for glucose-MS coupling"), shows that the Rabi splitting increases with the thickness of the glucose layer and eventually saturates once the thickness exceeds the plasmonic mode volume as characterized by the penetration depth $z_0$. The collective coupling can then be expressed as $g_N = g_0 \sqrt{\rho A z_0 \left(1 - e^{-2d_{gl}/z_0}\right)/2}$ where $g_0$ is the average coupling strength of a single emitter directly at the MS.

The measured intensity transmission of the coated MS is plotted in Fig. 3c. The transmission spectra of these measurements are obtained via time-resolved THz spectroscopy. The bare MS (black curve) initially

has a plasmonic resonance at 1.6 THz, corresponding to the narrow spectral dip in transmission. As we gradually deposit layers of glucose on the surface, we observe a redshift of the resonance to (1) 1.52 THz, (2) 1.49 THz, and (3) 1.43 THz (blue solid lines) with contributions of the Rabi splitting, leading to a maximum peak separation of 90 GHz at close to zero detuning. For design (3) in Fig. 3c, we calculate a FF of ~99% due to the strong overlap of the glucose region with the evanescent field of the MS (see SI, section Glucose filling fraction). The response of an equal-thickness coating of glucose on the substrate in an area with no MS elements is also plotted (blue dashed lines). The shift of the resonance can be understood as a gradual increase in the effective index of the medium on the top of the MS as we increase the thickness of the glucose layer[37]. Consistent with the theoretical prediction, we also have observed that further increasing the volume of glucose to exceed the plasmonic mode volume will no longer yield a redshift or an improvement in the Rabi splitting, but instead only increased absorption by glucose.

**Coupling with a standard Fabry–Perot cavity**

We systematically compare the performance of various cavity architectures shown in Fig. 4. The first row of Fig. 4 depicts, from left

to right, the FP cavity filled with glucose, the MS/planar mirror configuration of H1 with glucose on the planar mirror, and the MS/planar mirror configuration of H2 with glucose on the metasurface. The second row of Fig. 4 shows transmission maps of these three cavity designs obtained with the transfer matrix method to visualize the anti-crossing behavior indicative of the strong coupling regime. We use the relative change in cavity spacing, $\Delta d$, as a variable corresponding to the longitudinal displacement of one cavity mirror, which varies the air gap spacing between the two mirrors. The third row of Fig. 4 contains the corresponding experimentally extracted transmission maps showing good agreement with the model. The fourth row of Fig. 4 displays specifically selected transmission curves, including the zero-detuning cavity spacing, to highlight the polaritonic signatures.

Looking at the FP cavity architecture in Fig. 4a, one of the mirrors is coated with glucose using the aforementioned spray coating technique. THz-TDS measurements are taken at 5 µm increments of spacing between the cavity mirrors. This effective spacing is given by $d_{eff}(\omega) = d_{air} + d_{gl} n_{gl}(\omega)$, where $d_{air}$ is the air space between the mirrors, $n_{gl}(\omega)$ is the refractive index of glucose, and $d_{gl}$ is the thickness of the glucose layer. Analytically, one can deduce the expression for the transmission as

$$t_{FP}(\omega) = \frac{e^{-i\omega d_{eff}(\omega)/c}}{\zeta^2 + (1 - i\zeta)^2 e^{-2i\omega d_{eff}(\omega)/c}}, \quad (3)$$

where $\zeta$ describes the polarizability of the planar gold mirrors, which we assume to be frequency-independent. We find almost perfect agreement with the model and experimental measurements shown in the third row. The bright regions correspond to the cavity modes shifting in frequency with cavity spacing. An anti-crossing behavior is achieved around 1.43 THz, the bare resonance frequency of glucose when one of the cavity modes (of mode number $k = 4$) shifts from 1.6 to 1.3 THz with increasing $\Delta d$. We observe the formation of two characteristic polaritonic peaks in transmission, which are shown as two bright modes. A corresponding Rabi splitting of 140 GHz is observed. In the bottom row, we show the transmission of the mirror with a glucose coating (210 µm thick) and compare it to the transmission of the FP system at zero-detuning, which shows the symmetrical polaritonic splitting around the glucose transition. These results are consistent with prior work using a similar architecture with a pellet of lactose[18], noting that both the photonic and polaritonic modes appear as peaks in transmission.

For a standard FP cavity configuration, one would expect the field amplitude to decay completely at the planar mirror interface, and one would suspect that the molecular layer should be placed at the anti-nodes of the intracavity field. For this experiment, we calculate a FF of ~80%. Given the overall effective cavity spacing, the cavity field supports multiple modes and, thus, several anti-node positions. Therefore, coating a relatively thick molecular layer on one of the cavity mirrors provides a simple design approach to reach the strong light–matter regime with this architecture.

## Coupling with hybrid cavity architectures
While strong coupling can be achieved in both setups described above, integrating MSs with flat mirrors can bring a few advantages. One stems from the convolution of the MS's narrow frequency response of the MS with the cavity transmission window to design sharper resonances[34] and, thus, sharper polaritonic peaks. Another is the ease in cavity design complexity, allowing one to directly coat one of the cavity mirrors with a molecular/atomic ensemble rather than position a metasurface within a cavity field[38]; this advantage arises from the unique mode distribution and light–matter interaction enhancement which we explore in the subsequent section.

The plasmonic field of the MS peaks at the interface and decays exponentially with the distance (see SI, section "Response of plasmonic metasurface"). Therefore, a strong light–matter regime can be achieved when molecules with resonant transitions are deposited directly on the MS. In a hybrid cavity architecture using an MS as a reflective element, the field distribution also has a similar evanescent field distribution at that interface, which is demonstrated with numerical simulations in the subsequent section. Experimentally, we demonstrate that we can reach the strong light–matter coupling regime by filling up this plasmonic field volume with glucose in a hybrid cavity to obtain an even larger polaritonic splitting frequency than the one observed with a bare MS or FP resonator design. We also compare this hybrid cavity architecture to a similar one in which the glucose layer is deposited on the surface of the planar mirror instead of the MS reflector. Additionally, we compare this hybrid cavity architecture to a similar one in which the glucose layer is deposited on the surface of the planar mirror instead of the MS reflector and also the empty hybrid cavity without any glucose layer (see SI, section "Empty hybrid cavity").

The H1 architecture explored in Fig. 4b involves an uncoated MS with a resonance frequency of 1.45 THz merged with a glucose-coated flat mirror with a coating thickness of 210 µm (the same coated mirror used for the standard FP cavity experiments). In the transmission maps, one can see an anti-crossing region forming around the 1.43 THz vibrational resonance of glucose. Additionally, as a cavity mode ($k = 4$) shifts towards the MS/vibrational resonance, the linewidth of the cavity mode can be shown to narrow slightly. The bottom row of Fig. 4b compares the transmission characteristics of the H1 architecture when a cavity mode is overlapped with the MS/vibrational resonance (on-resonance, red curve) versus when a cavity mode is spectrally far from the MS/vibrational resonance (off-resonance, blue curve). We observe the formation of polariton peaks when the cavity mode is resonant with the MS/vibrational resonance. Since the glucose molecules within the H1 configuration are not within the plasmonic mode volume of the MS, the MS just acts as a frequency-selective mirror. Resultantly, the cavity mode dominates the photonic response of the hybrid cavity, and the frequency splitting and FF is the same as observed in the standard FP cavity arrangement, which is 140 GHz and ~80%, respectively.

The H2 architecture, with results shown in Fig. 4c, is particularly suited for the study of complex polaritonic systems as it couples the glucose resonance to both the cavity-delocalized mode and to the MS resonance. For this configuration, we incorporate the same coupled metasurface studied in Fig. 3 (coating layer (3)) into a hybrid cavity with an uncoated planar mirror. A coupled notch filter MS shows polaritons as minima in transmission, so resultantly, we observe the same transmission characteristics in this hybrid configuration since the glucose is still within the plasmonic mode volume of the metasurface, dominating the coupling interaction. First, when the cavity mode is spectrally far from the bare resonance of glucose, we find the same polaritonic splitting as observed with the MS/glucose architecture in Fig. 3. Then, as the cavity mode ($k = 4$) shifts towards the glucose resonance, the polaritonic splitting increases (SI, see section Constituent transmission curves for H2). Transmission spectra for the on- and off-resonant cases are plotted in the bottom row of Fig. 4c. We can observe how in the on-resonant case, the response of the H2 architecture shows an enhancement of the MS–glucose interaction, which leads to an effectively larger Rabi splitting of 200 GHz ($g_{eff} = 100$ GHz) for the MS–glucose polaritons. The exhibited splitting is now about a factor of two larger than the case of coupling only to the MS. Furthermore, with only 90 µm of glucose in this configuration, we calculate a FF of ~85%. The effect of the cavity mode is to enhance the Rabi splitting due to the additional field enhancement it provides. We discuss the origin of this polaritonic signature in more detail using numerical simulations in the subsequent section.

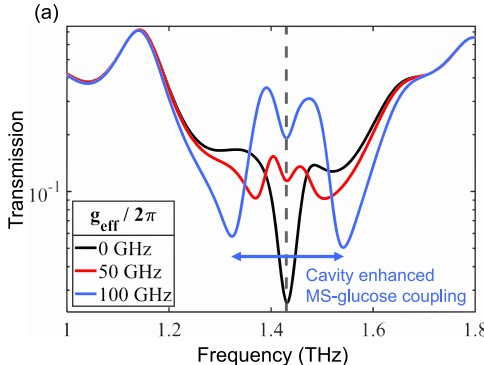

(a)

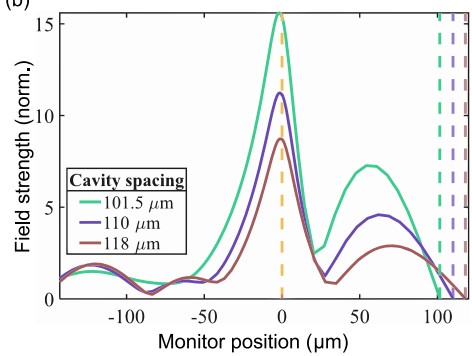

(b)

**Fig. 5 | Transfer matrix and FDTD simulations of the hybrid cavity. a** Transfer matrix simulation of cavity transmission (on-resonance) for different glucose−MS coupling strengths $g_{eff}$ and $d_{gl} = 100\,\mu m$. The arrows indicate the enhanced Rabi splitting of the MS−glucose interaction due to the cavity configuration. The vertical dashed line shows the location of the glucose resonance. When $g_{eff} > 0$, the glucose−MS polaritons are dominant, and the transmission result resembles experimental observations of the H2 configuration. Otherwise, when $g_{eff} = 0$, only the cavity−glucose interaction remains, and the transmission result resembles the H1 configuration. **b** A FDTD investigation of an empty hybrid cavity. The position of

the mirror, relative to the array interface ($0\,\mu m$, orange dashed line), determines the cavity spacing and, thus, the cavity resonance. Reducing the cavity spacing (from brown to purple to green) allows one to bring the cavity mode in resonance with the MS mode. The electric field magnitude within ($>0\,\mu m$) and outside ($<0\,\mu m$) of the cavity is monitored in one dimension and normalized to the incident field. The green curve shows the field profile obtained when the plasmonic and cavity modes are resonant with each other. The corresponding vertical dashed lines show the positions of the mirror. The enhancement of the field amplitude is consistent with the modified Rabi splitting showcased in (**a**).

## The nature of the hybrid cavity enhancement

To better understand the competition between the cavity−glucose and MS−glucose coupling for the H2 cavity architecture, we show in Fig. 5a a plot of the transfer matrix simulation results with increasing MS−glucose coupling strength $g_{eff}$ while keeping all other parameters fixed. When $g_{eff}$ is small, the cavity−glucose interaction can be seen from the formation of two weak polariton peaks symmetrically shifted from the dominant glucose resonance. In fact, when $g_{eff} = 0$, we retrieve the transmission response observed for the H1 architecture (black curve). When increasing the MS−glucose coupling, which is the case for H2 architecture, two dips appear, corresponding to the MS−glucose polaritons. Most importantly, the exhibited splitting is now roughly a factor of two larger than the case of coupling only to the MS. This enhancement can be traced back to the fact that the hybrid architecture leads to an increase in the zero-point electric field amplitude of the intracavity mode around the position where the glucose is added. To prove this point, we incorporate FDTD simulations, shown in Fig. 5b, of an empty hybrid cavity. The electric field strength is monitored in one-dimension along the MS−cavity axis and normalized to the incident field. The design consists of an infinitely periodic planar array of metallic crosses with a perfectly reflecting mirror plane at a distance (cavity spacing) above. In the plot, the orange dashed line indicates the position of the MS, and the green, purple, and brown dashed lines indicate the various positions of the mirror planes. Three different cavity spacings are compared to showcase when a cavity mode is completely overlapped (green curve), partially overlapped (purple curve), and not overlapped (brown curve) with a cavity mode. The electric field is monitored at the resonance frequency of the plasmonic array. Positive monitor positions are within the cavity, and negative monitor positions are outside of the cavity. The resultant field profiles obtained from these simulations clearly show that the field strength at the MS interface of a hybrid cavity can be significantly enhanced (for example, a factor around 1.8 close to the interface and even larger further away from it) when a cavity mode is resonant to a plasmonic mode. As aforementioned, the coupling strength, $g$, is linearly proportional to the local electric field given by the product of the zero-point electric field amplitude and the spatial function of the mode supported by the photonic resonator.

Going beyond the approach of using transfer matrix calculations and FDTD simulations, one can also formulate a quantum description of the hybrid cavity enhancement. The intracavity field of a hybrid

cavity architecture can be modeled by considering a coupling interaction between the MS and the planar cavity mode. The Hamiltonian for this system is

$$H = \omega_a \hat{a}^\dagger \hat{a} + \omega_d \hat{d}^\dagger \hat{d} + \lambda \left( \hat{a}^\dagger + \hat{a} \right) \left( \hat{d}^\dagger + \hat{d} \right), \qquad (4)$$

where $\hat{a}$ represents the planar cavity mode, $\hat{d}$ represents the MS mode, and $\lambda$ is a coupling parameter[34]. The interaction described by Eq. (4) yields a new hybrid photonic mode, which is characterized by a single peak in transmission. When glucose is introduced into this architecture, one can then use the hybrid photonic mode in the Tavis−Cummings Hamiltonian of Eq. (1) to describe the coupled system. The coupling strength is then partially determined by the spatial overlap between the glucose and the hybrid field distribution.

In summary, we have shown a strong coupling between a vibrational resonance of glucose and a plasmonic MS mode in the THz regime, where powerful spectroscopic methods based on the THz-TDS technique can be exploited for system characterization. Furthermore, we suggest that this work can open avenues in designing light−matter interfaces based on hybrid architectures. We have shown experimentally, and with support from analytics tested against numerical simulations, the emergence of strong light−matter coupling in a variety of geometries, ranging from the standard FP cavity to more complex hybrid configurations interfacing an MS with a planar mirror. We have demonstrated the enhancement of light−matter coupling strength brought on by the hybrid cavity design and directly connected it to the increase in the zero-point electric field amplitude stemming from the interference between the MS evanescent field and the standing wave field of the cavity at the location of the glucose layer. Further investigations in this direction hold the promise of identifying mechanisms for the design of higher-finesse cavities as well as providing platforms for the exploration of the richer physics of multi-polariton systems.

## Methods
### Terahertz time-domain spectroscopy
All experimental measurements taken for this work in the THz region are accomplished with a time-domain spectroscopy technique (THz-TDS) using the setup depicted in Fig. 2. Briefly, an ultrafast source operating at 1030 nm with a pulse duration of 170 fs and a repetition

rate of 1.1 MHz is split into a pump and gating beams. The pump generates a THz transient through optical rectification in a 2 mm-thick GaP crystal. A standard electro-optic sampling scheme resolves the THz pulse using another 2 mm-thick GaP crystal. The THz-induced birefringence, proportional to the local THz field, is resolved with polarization optics (a quarter wave plate, a Wollaston prism, and a pair of balanced photodiodes) as a translational stage changes the relative time delay between the two pulses. As a result, the full oscillating THz field can be resolved.

## Metasurface design and fabrication

MS designs utilized for this work were fabricated with a photo-lithography process using a photomask. A 150 nm-thick aluminum cross array was deposited atop a 188 μm-thick cyclo-olefin copolymer, commercially branded as Zeonor, with a measured index of refraction of 1.53 (at 1 THz). The Zeonor substrate was selected due to its dispersionless quality and THz transparency. The photolithography procedure utilizes a negative tone resist to etch the array design, followed by an aluminum deposition through direct current sputtering and a subsequent lift-off process. Two MS designs are shown in the results of this work and defined in terms of the cross periodicity ($P$), arm length ($L$), and arm width ($W$). The glucose coated MS, with a coating thickness of $(90 \pm 10)$ μm, utilized for the results of Fig. 3 and H2 of Fig. 4 is defined by the geometry $P/L/W = 110.25/63/9.54$ μm and has a resonance frequency of 1.6 THz. For the results of H1 of Fig. 4 in the main text, an MS with geometry $P/L/W = 124.25/71/10.76$ μm, with a resonance frequency at 1.45 THz, is fabricated.

## Planar cavity design

Planar mirror samples for the cavity architectures described in this work were fabricated by sputtering 9 nm of gold on a 650 μm-thick undoped GaAs substrate with a refractive index of 3.6 (at 1 THz). One mirror sample was then coated with a $(210 \pm 10)$ μm-thick glucose layer and utilized for the FP and H1 cavity architectures shown in Fig. 4. Another mirror sample that was not coated is utilized to produce the FP and H2 cavity architectures shown in Fig. 4.

## Glucose spray coating technique

α-D-Glucose was acquired (MilliporeSigma) in a solid-state powder form. The THz refractive index of glucose was measured from a pellet sample created utilizing a hydraulic press to achieve a pellet diameter of 5 mm with a 300 μm thickness (see SI, section "Glucose spray coating"). The absorption of this sample is shown in Fig. 2 and used to characterize glucose. While pellets are commonly used in standard FP cavity experiments, we found that they lacked robustness for experiments involving MSs due to pellet fragility, thickness/diameter scaling, and the inability to ensure a strong concentration of glucose between the plasmonic elements of our arrays. We instead use a spray coating technique to place a large concentration of glucose molecules at the interface of the plasmonic array, depicted in Fig. 3a. The vibrational resonances of glucose, observed at 1.43 and 2.09 THz, are prominent when glucose is in solid-state form. All glucose-coated MSs and planar mirrors produced for this work were coated using this spray coating technique. Placing many solid-state glucose molecules near the plasmonic interface of the MS requires the crystalline particle size of glucose to be reduced; we achieve this condition utilizing an ultrasonic bath. First, we prepare a mixture of glucose powder in isopropanol, which is non-polar and has a low evaporation temperature. The mixture is then sonicated to produce a stable suspension of glucose in isopropanol with crystalline sizes of ~10 μm. The extracted suspension of fine glucose particulates is loaded into an air spray gun. The sample to be coated is placed onto a hot plate with a temperature set above the boiling point of isopropanol (82.5 °C) to facilitate the rapid evaporation of the solvent. A few sprays of the mixture onto the substrate

and subsequent drying results in a relatively uniform layer of solid-state glucose on the substrate. To increase the layer thickness, the cycle of spray coating and drying is repeated. The thickness of spray-coated glucose onto the samples is probed using optical microscopy.

## Data availability

The processed transmission datasets are available at https://doi.org/10.6084/m9.figshare.25414015.

## Code availability

The code used for this study is available upon reasonable request from the corresponding authors.

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

## Acknowledgements

K.D., R.W.B., and J.-M.M. acknowledge funding from the Natural Sciences and Engineering Research Council of Canada (NSERC) Strategic Partnership Program (STPGP/ 521619-2018). J.M.M. acknowledges the NSERC Discovery funding program (RGPIN-2016-04797, RGPIN-2023-05365), the Canada Foundation for Innovation (CFI) (Project Number 35269), and the Ontario Ministry of Colleges and Universities' Early Researcher Award (ER21-16-206). C.G. and M.R. acknowledge financial support from the Max Planck Society and the Deutsche Forschungsgemeinschaft (DFG, German Research Foundation)—Project ID —429529648—TRR 306 QuCoLiMa ("Quantum Cooperativity of Light and Matter").

## Author contributions

A.J., M.R., A.S., A.M., C.G. and J.-M.M. conceptualized the idea and plan for this work. A.J. completed the FDTD simulations for the metasurfaces, and Y.X. fabricated the metasurfaces. A.S. fabricated the mirrored samples and applied the spray coating technique to all samples. M.R., C.G., J.-M.M. and A.J. contributed to the development and application of the theory. A.J. carried out the measurements. A.J., J.-M.M. and A.S. analyzed the experimental results. K.D., R.W.B., B.T.S., C.G. and J.-M.M. supervised this work.

## Competing interests

The authors declare no competing interests.
