## [Peer Review File · Nature Communications]

Hybrid architectures for terahertz molecular polaritonicsREVIEWER COMMENTS

Reviewer #1 (Remarks to the Author):

The manuscript by Jaber et al. studies strong light-matter coupling in the THz domain between intermolecular vibrations in glucose and hybrid cavities incorporating metasurfaces as one of the mirrors. The authors present experimental results corroborated by theoretical modeling and numerical simulations, focusing on the linear spectral response of these coupled architectures. Most notably, the authors demonstrate that such hybrid cavity-metasurface structures can enhance the interaction strength with the material (i.e. vibrational excitations in the glucose crystals).

Generally, I find this work both interesting and timely, as strong coupling at terahertz frequencies in various material systems is gaining increasing attention. While the current study does contribute to advancing the field and is potentially relevant to a broad readership, the authors may be aware of a closely related study presented in *Nano Lett.* 2021, 21, p. 1320, which reported similar enhancement in a hybrid FP/metasurface system under strong coupling with mid-IR vibrational modes. The current manuscript studies these effects at the terahertz region, but in my opinion, it does not have the required novelty to justify publication in *Nature Comm.* but rather in a more specialized journal.

Apart from that, the authors may consider to address the following points:

1. In the introduction the authors state that "In the THz regime (wavelengths from 0.1 to 1 mm), light can also directly couple with intra- and inter-molecular vibrational modes [10–12]". However, intramolecular vibrations are usually associated with the mid-infrared regime (~3-30 microns). Refs. 10, 11 and 13-15 involve molecular vibrations exactly at this spectral range.
2. On page 2, the authors write that they demonstrate similar performances of the metasurface and FP structure, but then say that "This is due to the stronger spatial confinement of a photonic mode allowed by the evanescent field of the MS which enables a reduction in the number of molecular emitters". I find this phrasing somewhat confusing. If the confinement is stronger, why do similar performances are observed?
3. In Fig. 4, does "cavity spacing" refers to the distance between the mirrors or the thickness of the air gap?
4. On page 6, the authors write "one of the cavity modes displays an anti-crossing behavior ...". Which one is it?
5. The term "density plot" might be misleading, since Fig. 4 does not correspond to density. I suggest the authors use a more common term to refer to their false-color images.
6. When discussing the results in Fig. 4(a) the Rabi splitting is measured as the distance between two transmission peaks, while in 4(c) the polaritons are associated with transmission dips (and in 4(b) it is not clear how the splitting is defined). I understand that generally in different structures resonances can appear in different manners (transmission maximum/minimum). Still, perhaps a safer way to identify the polariton frequencies would be to simulate the total absorption, where the polaritons should always give rise to maxima. I believe that this will also provide better assessments of the enhancement in the interaction strength observed for the H2 sample.
7. It is not clear whether the three architectures have the same filling fraction (i.e. ratio between the volume of the glucose and the total volume between the mirrors). I believe that comparing these filling fractions is essential when claiming that the interaction strength is enhanced.

Reviewer #2 (Remarks to the Author):

In their manuscript titled "Hybrid architectures for molecular polaritonics" by Jean-Michel Ménéard et al, the authors report fabrication, simulation and experimental characterization of what they call hybrid architectures to enhance light-matter coupling between dipole resonances in complex molecules and light at terahertz (THz) frequencies.

I like this work, but in many respects, it needs to be substantially improved to meet the publication standards set forth by Nature family of journals. With sufficient reworking of the manuscript, I hope the authors can improve the quality to an adequate level. I have for you below an itemized summary of my feedback that guides my decision.

- The novelty of this work needs to be adequately motivated. It is not clear to a non-specialist in this field whether the metasurface approach used for mirrors in (molecular) polaritonics has been tried in prior works – it seems like a straightforward idea, especially at low frequencies, where metasurface fabrication tolerances are pretty relaxed. If this was explored by others, I failed to see a discussion dedicated to the crystallization of the novelty of this work in the context of the prior reports. The authors remark that THz range is less explored, but then the paper's focus should be narrowed down to that range, including already in the title (which is, at present, too broad and hence is in effect over-claiming the anticipated impact).

- One of the parameters of light-matter coupling should be the overlap of the photonic density of states with the position of the intracavity layer containing the matter part. This is not explained at all, however both the thickness of the matter layer and its position can drastically affect the overall coupling to a standing photonic wave. A naïve expectation would be that the optical field magnitude at a position of a (perfect) mirror should be zero, hence placing a layer near the cavity boundaries would not be a textbook choice. Presented simulations, shown toward the end of the manuscript, seem to disprove this naïve picture, however no discussion is given. The effect is likely due to the complex response function of the MS and, in fact, tunability of these effects could have been elucidated as one of the advanced approaches for one would use MS in the "hybrid" designs.

- THz measurements here are reported as "time-resolved", however this can be misleading to a non-specialist audience. Instead, EOS here is used instead of an FTIR or any other standard spectroscopic analysis. It is not clear why the EOS measurement is used here, is it due to better SNR compared to say FTIR? What is the frequency range over which EOS is operating? Since the technique measures both real and imaginary components of the electric field, why is the phase response (dispersion of the refractive index) of the sample not shown in the main text? Is Fig1 of the supplementary showing this? If it is (not clear from the current narrative) – make sure you properly reference figures/results in the Supplementary section in the main text or better yet transfer this to the main part and draw more emphasis to this, effectively justifying EOS also. (Aside: n_g notation is typically reserved for the group refractive index, so it might be confusing when n_g is quoted in the text in a different context. Perhaps the ambiguity can be omitted with "phase refractive index of glucose (n_g) is given by ...".)

- My understanding of the critical aspect of the hybrid design is the inclusion of a resonant effect in the frequency response of the metasurface (even though I fail to see a clear justification for this). But my main concern for the presentation is why the MS's resonance hasn't been measured in the same manner as that of the glucose layer. It would have been very instructive to compare both responses of the "bare" cavity, signal that contains only MS resonances, with that of the "loaded" cavity, now containing the anti-crossing behavior in the hybridized modes. Some version of that is shown in Fig 2, but again, the response of the MS (important for this discussion) hasn't been properly isolated.

- In the presented form of the Hamiltonian, the importance of the angle formed between the polarization of light and the orientation of the dipole is briefly mentioned. However, for a complete and balanced presentation, one needs to include a discussion on whether a specific orientation of the dipole moment of the glucose layer has been ensured (specifics of fabrication and/or post-processing?) and verified. This is an important step. Suppose the glucose response is isotropic:

Why couldn't the effect be demonstrated more clearly by switching the polarization of light (using the 90deg rotation symmetry of the MS) from aligned to perpendicular orientation with the dipole?

- Simulation results have been mentioned several times in the main text, however, it is surprising that the main experimental finding reported in Figure 2 (where the 200 GHz splitting is presented) is not shown alongside the numerics. It would strongly add to the clarity and credibility of the findings. In particular, the appearance of the 3rd peak in the main result showing 200 GHz anticrossing is not discussed in much detail. It is indeed an interesting feature and I would have liked to see a creative/insightful discussion about how can this feature (due to MS, correct?) be used to improve/advance/enhance understanding/applications of polaritonics.

The authors demonstrate strong coupling regime between THz vibration transition of glucose (1.5 THz) and a hybrid photonic architecture, which consists of a Fabry-Perot cavity where one of the mirrors is replaced with a meta-surface. The authors compare structures which consist of meta-surface alone, Fabry-Perot cavity alone, and a hybrid cavity, study the light-matter coupling constant which can be inferred from the optical spectroscopy of the system. Namely, the coupling strength is proportional to the spectral anti-crossing of the light-matter coupled (polariton) modes. The authors claim strong enhancement of the light-matter coupling constant in a hybrid cavity. As shown further, I strongly contest this result which seem to be a misinterpretation of the experimental data, as commented further.

Before detailing this point, I would like to point out that the phenomenon of strong coupling between THz vibrational transitions has been already been observed (<https://doi.org/10.1038/s41467-019-11130-y>). Also, many THz hybrid photonic architectures have already been demonstrated (<https://doi.org/10.1021/acsp Photonics.1c00717>, <https://doi.org/10.1364/PRJ.482195>, <https://doi.org/10.1364/OE.456044>). In particular, the last two references present a much more advanced analysis of hybrid structures than the one proposed by the authors (see Fig.5 and the corresponding discussion). Thus, there is a serious doubt that the paper contains sufficient novelty to grant a publication at Nature Communications.

Figure R1.

Next, I contest the author's interpretation of the polariton splitting. This is explained in the Figure above. In Figure (c) lower panel, there are actually 3 resonant features that can be attributed to polariton states (LP: lower polariton, UP: upper polariton). Comparing between the lower and middle panel of Figure (c), one can conclude that: a) the spectrum chosen by the authors does not actually correspond to the anti-crossing point; b) the UP is thus wrongly attributed to an adjacent photonic

mode by the authors, and the UP polariton is actually the feature at much lower frequency as pointed out in red. Thus, the polariton splitting is much lower than the one claimed by the authors; actually a closer look at the middle panels in the Figure reveals that the polariton splitting is essentially the same in all three cases. Note that the analysis is rather complicated by the fact that the polariton anti-crossing is on the same order of magnitude (even lower) than the linewidth of the resonances. The best way to determine the polariton splitting would be to plot the positions of all observed resonances as a function of the cavity spacing that can be extracted from the middle panels. Such graph, combined a line indicating the 1.5 absorption of glucoses would allow to determine precisely the cavity spacing for which the anti-crossing appears.

We thank the Reviewers for their extremely constructive feedback on our manuscript, which allowed us to offer a better highlight of the novelty of our hybrid architecture design and to improve the clarity of our presentation. We have addressed each Reviewer's remark on a comment-by-comment basis and we have made appropriate modifications to the text and figures of our manuscript. Overall, we believe that the comments greatly helped us to improve the manuscript and highlight the novelty of our approach.

In this document, the Reviewer's comments are in **black**, our responses are in **blue**, and all changes to the manuscript are in **red**.

Reviewer 1:

The manuscript by Jaber et al. studies strong light-matter coupling in the THz domain between intermolecular vibrations in glucose and hybrid cavities incorporating metasurfaces as one of the mirrors. The authors present experimental results corroborated by theoretical modeling and numerical simulations, focusing on the linear spectral response of these coupled architectures. Most notably, the authors demonstrate that such hybrid cavity-metasurface structures can enhance the interaction strength with the material (i.e. vibrational excitations in the glucose crystals).

Generally, I find this work both interesting and timely, as strong coupling at terahertz frequencies in various material systems is gaining increasing attention. While the current study does contribute to advancing the field and is potentially relevant to a broad readership, the authors may be aware of a closely related study presented in Nano Lett. 2021, 21, p. 1320, which reported similar enhancement in a hybrid FP/metasurface system under strong coupling with mid-IR vibrational modes. The current manuscript studies these effects at the terahertz region, but in my opinion, it does not have the required novelty to justify publication in Nature Comm. but rather in a more specialized journal.

R1-0: We thank the Reviewer for finding our work “interesting and timely” and recognizing that our study “does contribute to advancing the field and is potentially relevant to a broad readership”. We also thank the Reviewer for mentioning the Nano letters work. We believe this is a relevant work exploring molecular polaritonics in the mid-infrared region. This region, however, has been explored by many research groups in the context of molecular polaritonics, while very few of these experimental results in the THz region have been published, potentially because the coupling to molecules is more complex since it may involve a mix of molecular and crystal resonances. This is one novelty aspect of our work, as already mentioned by the Reviewer. But more importantly, we investigate a new type of hybrid cavity where a reflective plasmonic array replaces one of the mirrors of a Fabry-Perot (FP) cavity to achieve an uncharted electromagnetic mode distribution. This is a fundamentally different design from the one reported in this Nano Lett. and in similar work, e.g. R. Amelin and H. Giessen. Nano Lett., 10, 4394-4398 (2010) and D. G. Baranov et al., Nat Commun., 11, 2715 (2020), which relies on a plasmonic array inserted inside a standard FP cavity. The novelty of our design covers both the architecture aspect and related applications to molecular polaritonics: With the use of a reflective metasurface (MS) as one of the cavity mirrors, we can deposit a layer of molecules directly at the interface of the MS and achieve optimal light-matter interaction. We use this unique design to increase the Rabi splitting frequency by 40% in comparison to the one obtained with the FP or MS resonator alone. We also synergize our experimental work with numerical results to provide insight into the rich physics of these new hybrid architectures. Molecular polaritonics in the THz regime is a growing field of research and applications, as the Reviewer points out, and we believe that progress in designing architectures for enhanced light-matter coupling is a crucial building block for future vibrational polaritonic studies, thus allowing new chemical and biological applications.

We made significant changes to the manuscript to clarify the novelty aspects of our work. We now also discuss the class of papers mentioned by the Reviewer:

[P. 1] Within regions of lower energies within the electromagnetic spectrum, light can also directly couple to intra- and inter-molecular vibrational modes¹⁰⁻¹². This can lead to emerging applications in chemistry through the modification of ground state potential landscapes allowing for the manipulation of chemical reaction pathways and rates¹³⁻¹⁵. However, for the purpose of studying light-matter coupling involving far-infrared light and low-energy vibrational transitions of molecules, the DBR design becomes unpractical since most dielectrics are absorptive in this region and their required thickness exceeds the capacity of most nanofabrication equipment. In the THz region (wavelengths from 0.1 to 1 mm), the coupling to low energy bonds, such as intermolecular vibrations or hydrogen bonding, becomes accessible^{16,17}. At these larger

wavelengths, Fabry-Perot (FP) cavities utilizing metallic planar mirrors have been demonstrated, where a VRS on the order of 68 GHz has been reached¹⁸. Following this framework, more sophisticated cavity architectures can potentially lead to even stronger polaritonic splitting with molecular systems in the THz region.

In the context of light-matter interaction with solid-state emitters, different types of THz resonators based on plasmonic emitters/antennas^{19,20}, waveguides^{21,22}, and metasurfaces (MSs)²³⁻²⁵, or a combination of them²⁶⁻²⁹, have been explored to achieve a strong field confinement. However, THz cavity architectures involving multiple electromagnetic resonators have not yet been explored in combination with resonant molecular ensembles, although few of these structures optimized for the mid-infrared range have been investigated^{30,31}. Thus, a systematic study of metasurface-based cavities in the emerging context of THz molecular polaritonics is needed to enable a range of chemistry applications such as those discussed above.

Here we propose a set of four THz resonator architectures illustrated in Fig. 1, including two hybrid designs relying on a reflective MS (analogous e.g., to subradiant optical mirrors realized with trapped atoms³²) replacing the planar mirror of a FP cavity to achieve better cavity finesse³³. These designs are used to yield unprecedented electromagnetic field spatial distributions peaking at the interface of the MS. We explore the coupling of these fields to a resonant transition of glucose, a molecule with a strong relevance to biological processes including metabolism and photosynthesis. Until now, the coupling of molecules with hybrid cavities has remained mostly unexplored, in part due to the complexity of fabricating a dense layer of organic material within a spatially confined electromagnetic mode volume. Using a spray coating technique³⁴, we obtain a dense molecular layer in direct contact with the surface of the MS allowing us to reach the strong light-matter coupling regime involving a resonant plasmonic and molecular vibrational mode. This system can then be inserted in the design of a planar resonator geometry to enable further hybridization with a photonic mode.

We experimentally and numerically compare the transmission properties of such cavity architectures, exploring the evolution from a standalone resonator towards a hybrid design with and without (see Supplementary Information (SI), section *Empty hybrid cavity*) the presence of a resonant molecular transition. This systematic experimental study supported by theory and simulations, provides a unique window into the complex interaction between plasmonic, photonic and vibrational resonances. Most notably, we compare the polaritonic splitting achieved with a plasmonic MS and a standard FP cavity design. Interestingly, a similar Rabi splitting is observed although fewer molecular emitters can be contained within the evanescent mode volume of the MS, in comparison to those within the FP cavity mode. This indicates that the metasurface, through a larger mode confinement and stronger field enhancement, enables an enhanced coupling per molecule. Then, using a hybrid cavity architecture, where the MS replaces one of the mirrors of the FP cavity, we reach a 40% larger polaritonic splitting with less glucose volume than a standard FP due to a modified intracavity field.

Apart from that, the authors may consider to address the following points:

1. In the introduction the authors state that “In the THz regime (wavelengths from 0.1 to 1 mm), light can also directly couple with intra- and inter-molecular vibrational modes [10–12]”. However, intramolecular vibrations are usually associated with the mid-infrared regime (~3-30 microns). Refs. 10, 11 and 13-15 involve molecular vibrations exactly at this spectral range.

R1-1: We thank the Reviewer for this comment. We have made this correction by properly distinguishing the type of transitions involving vibrational transitions in the mid-infrared and THz regimes. We also added relevant references to THz intermolecular vibrations in glucose:

[P. 1] Within regions of lower energies within the electromagnetic spectrum, light can also directly couple to intra- and inter-molecular vibrational modes¹⁰⁻¹². [...] In the THz region (wavelengths from 0.1 to 1 mm), the coupling to low energy bonds, such as intermolecular vibrations or hydrogen bonding, becomes accessible^{16,17}.

2. On page 2, the authors write that they demonstrate similar performances of the metasurface and FP structure, but then say that “This is due to the stronger spatial confinement of a photonic mode allowed by the evanescent field of the MS which enables a reduction in the number of molecular emitters”. I find this phrasing somewhat confusing. If the confinement is stronger, why do similar performances are observed?

R1-2: We agree with the Reviewer and have now modified this section:

[P. 3] Most notably, we compare the polaritonic splitting achieved with a plasmonic MS and a standard FP cavity design. Interestingly, a similar Rabi splitting is observed although fewer molecular emitters can be contained within the evanescent mode volume of the MS, in comparison to those within the FP cavity mode. This indicates that the metasurface, through a larger mode confinement and stronger field enhancement, enables an enhanced coupling per molecule.

3. In Fig. 4, does “cavity spacing” refers to the distance between the mirrors or the thickness of the air gap?

R1-3: “Cavity spacing” indicates the displacement of one of the reflectors of the cavity relative to an initial position. As we increase the cavity spacing, we increase the air gap inside the cavity. To prevent any confusion, we now refer to this value as Δd (μm): the variation of the cavity spacing:

[P. 7] The second row of Fig. 4 shows transmission maps of these three cavity designs obtained with the transfer matrix method to visualize the anti-crossing behaviour indicative of the strong coupling regime. We use the relative change in cavity spacing, Δd , as a variable corresponding to the longitudinal displacement of one cavity mirror, which varies the air gap spacing between the two mirrors.

4. On page 6, the authors write “one of the cavity modes displays an anti-crossing behavior ...”. Which one is it?

R1-4: We thank the Reviewer for pointing out this issue. We have now reworded the sentence to clearly indicate what mode is displaying the anti-crossing behavior. We have also added a white dashed line over the colour figures to indicate the frequency of the bare transition in glucose contributing to light-matter interactions:

[P. 7] The bright regions correspond to the cavity modes shifting in frequency with cavity spacing. An anti-crossing behavior is achieved around 1.43 THz, the bare resonance frequency of glucose, when one of the cavity modes shifts from 1.6 to 1.3 THz with increasing Δd . We observe the formation of two characteristic polaritonic peaks in transmission which are shown as two bright modes.

5. The term “density plot” might be misleading, since Fig. 4 does not correspond to density. I suggest the authors use a more common term to refer to their false-color images.

R1-5: We now changed the term “density plot” to “**transmission map**”, which is used in the same context in previous works such as M. Hertzog et al. Nano Lett., 21, 1320-1326 (2021) .

6. When discussing the results in Fig. 4(a) the Rabi splitting is measured as the distance between two transmission peaks, while in 4(c) the polaritons are associated with transmission dips (and in 4(b) it is not clear how the splitting is defined). I understand that generally in different structures resonances can appear in different manners (transmission maximum/minimum). Still, perhaps a safer way to identify the polariton frequencies would be to simulate the total absorption, where the polaritons should always give rise to maxima. I believe that this will also provide better assessments of the enhancement in the interaction strength observed for the H2 sample.

R1-6: We now provide some clarifications on the nature of the transmission spectra observed for different configurations (see explanations and modifications to the manuscript below).

We also thank the Reviewer for their suggestion to look at the absorption spectra, which is indeed an interesting one. Our conclusion, after taking into account the behavior of absorption features in the polaritonic system is that a few particularities of the system make absorption an untrustworthy witness of strong coupling. In order to understand this aspect, we have simulated the simplest possible example, where both a cavity quantum electrodynamics formulation (based on the Jaynes-Cummings Hamiltonian) and a standard transfer matrix approach give identical results. In the standard quantum optics approach [D. Plankensteiner, et al., Phys. Rev. A 99, 043843 (2019)], the proper definition of strong coupling is connected to the emergence of coherent light-matter interactions marked by the occurrence of a few Rabi oscillations before losses take over. This can be seen to coincide with the emergence of separable peaks in transmission but not necessarily with distinguishable peaks in absorption. Mathematically, the condition one asks is that a Hopf bifurcation occurs,

which imposes the following inequality between the coherent coupling (g) and the losses in the system:

$$g > \frac{|\kappa - \gamma|}{2}$$

where κ is the mirror loss and γ is the linewidth of the quantum system. Fulfilling this condition will result in polariton splitting that allows the coupled system to be modelled as a sum of two separated Lorentzians ($2g$ distance) which are centered around the upper and lower polariton frequencies. For this simplified system, we plot the transmission, reflection, and absorption below:

Fig. R1-6: Transmission, reflection and absorption of a strongly coupled light-matter system under the Jaynes-Cummings model for a scan of the cavity input field around the cavity resonance.

The x-axis is the frequency detuning from the matter transition normalized to the mirror loss. Like our experiment, we set the mirror and radiative losses to just allow the strong coupling condition to be met and we see clear split peaks in transmission and reflection. In absorption, even though we only consider mirror and radiative losses, we cannot reasonably resolve the polaritonic features.

Furthermore, in a system like our experimental setup, additional signatures of absorption at the bare matter frequency will occur, as part of the matter system lies outside the confined field region when the glucose layer is increased. This makes the reading out of absorption spectra extremely complex [F. Stete, et al., ACS Photonics, 4, 1669–1676 (2017)].

Regarding the nature of the polaritons (peaks or dips) in Figs. 4 (a-c):

4(a). This is the standard cavity-matter coupling experiment. Our results showing both polaritons as transmission peaks symmetrically shifted from the bare resonance frequency are similar to the results obtained with lactose in a Fabry-Perot cavity at THz frequencies (R. Damari, et al., Nat Commun., 10, 3248 (2019)).

4(b). Since the glucose within this configuration (H1) is not within the plasmonic mode volume of the metasurface, the metasurface just acts as a frequency selective mirror. Having a high reflectance at the glucose molecular transition frequency results in an overlapping cavity mode and thus the same overall picture as the one observed in a standard Fabry-Perot cavity design where polaritons appear as maxima in transmission.

4(c). This picture (H2) corresponds to the same system studied in Fig. 3 where polaritons show up as a minimum in transmission. We observe the same behavior here since the glucose is within the plasmonic mode volume of the metasurface. The effect of the Fabry-Perot cavity is to enhance the Rabi splitting due to the additional field enhancement it provides. We discuss the origin of this polaritonic signature in more details using numerical simulations, e.g. see Fig. 5.

In the relevant sections on *Coupling with a standard Fabry-Perot cavity* and *Coupling with hybrid cavity architectures*, we have made modifications to the manuscript:

Coupling with a standard Fabry-Perot cavity. We systematically compare the performance of various cavity architectures shown in Fig. 4. The first row of Fig. 4 depicts, from left to right, the FP cavity filled with glucose, the MS/planar mirror configuration of H1 with glucose on the planar mirror, and the MS/planar mirror

configuration of H2 with glucose on the metasurface. The second row of Fig. 4 shows transmission maps of these three cavity designs obtained with the transfer matrix method to visualize the anti-crossing behaviour indicative of the strong coupling regime. We use the relative change in cavity spacing, Δd , as a variable corresponding to the longitudinal displacement of one cavity mirror, which varies the air gap spacing between the two mirrors. The third row of Fig. 4 contains the corresponding experimentally extracted transmission maps showing good agreement with the model. The fourth row of Fig. 4 displays specifically selected transmission curves, including the zero-detuning cavity spacing, to highlight the polaritonic signatures.

Looking at the FP cavity architecture in Fig. 4 (a), one of the mirrors is coated with glucose using the aforementioned spray coating technique. THz-TDS measurements are taken at 5 μm increments of spacing between the cavity mirrors. This effective spacing is given by $d_{\text{eff}}(\omega) = d_{\text{air}} + d_{\text{gl}}n_{\text{gl}}(\omega)$, where d_{air} is the air space between the mirrors, $n_{\text{gl}}(\omega)$ is the refractive index of glucose, and d_{gl} is the thickness of the glucose layer. Analytically, one can deduce the expression for the transmission as

$$t_{FP}(\omega) = \frac{e^{-i\omega d_{\text{eff}}(\omega)/c}}{\zeta^2 + (1-i\zeta)^2 e^{-2i\omega d_{\text{eff}}(\omega)/c}}, \quad (3)$$

where ζ describes the polarizability of the planar gold mirrors, which we assume to be frequency-independent. We find almost perfect agreement with the model and experimental measurements shown on the third row. The bright regions correspond to the cavity modes shifting in frequency with cavity spacing. An anti-crossing behavior is achieved around 1.43 THz, the bare resonance frequency of glucose, when one of the cavity modes shifts from 1.6 to 1.3 THz with increasing Δd . We observe the formation of two characteristic polaritonic peaks in transmission which are shown as two bright modes. A corresponding Rabi splitting of 140 GHz is observed. In the bottom row, we show the transmission of the mirror with a glucose coating (210 μm thick) and compare it to the transmission of the FP system at zero-detuning, which shows the symmetrical polaritonic splitting around the glucose transition. These results are consistent with prior work using a similar architecture with a pellet of lactose¹⁸.

For a standard FP cavity configuration, one would expect the field amplitude to decay completely at the planar mirror interface and one would suspect that the molecular layer should be placed at the anti-nodes of the intracavity field. For this experiment, we estimate a glucose *FF* of $\sim 80\%$. Given the overall effective cavity spacing, the cavity field supports multiple modes and thus several anti-node positions. Therefore, coating a relatively thick molecular layer on one of the cavity mirrors provides a simple design approach to reach the strong light-matter regime with this architecture.

Coupling with hybrid cavity architectures. While strong coupling can be achieved in both setups described above, integrating MSs with flat mirrors can bring a few advantages. One stems from the convolution of the MS's narrow frequency response with the cavity transmission window to design sharper resonances³³ and thus sharper polaritonic peaks. Another arises from the interaction of three resonances involving the glucose, MS and cavity, to yield a potentially richer polaritonic physics beyond the standard upper and lower polariton states typical of strong coupling experiments. An additional advantage is the ease in cavity design complexity allowing one to directly coat one of the cavity mirrors with a molecular/atomic ensemble rather than position a metasurface within a cavity field³⁸; this advantage arises from the unique mode distribution and light-matter interaction enhancement which we explore in the subsequent section.

The plasmonic field of the MS peaks at the interface and decays exponentially with the distance (see SI, section *Response of a plasmonic metasurface*). Therefore, a strong light-matter regime can be achieved when molecules, with resonant transitions, are deposited directly on the MS. In a hybrid cavity architecture using a MS as a reflective element, the field distribution also has a similar evanescent field distribution at that interface, which is demonstrated with numerical simulations in the subsequent section. Experimentally, we demonstrate that we can reach the strong light-matter coupling regime by filling up this plasmonic field volume with glucose in a hybrid cavity, to obtain an even larger polaritonic splitting frequency than the one observed with a bare MS or FP resonator design. We also compare this hybrid cavity architecture to a similar one in which the glucose layer is deposited on the surface of the planar mirror instead of the MS reflector. Additionally, we compare this hybrid cavity architecture to a similar one in which the glucose layer is deposited on the surface of the planar mirror instead of the MS reflector and also the empty hybrid cavity without any glucose layer (see SI, section *Empty hybrid cavity*).

The H1 architecture explored in Fig. 4(b) involves an uncoated MS with a resonance frequency of 1.45 THz merged with a glucose coated flat mirror with a coating thickness of 210 μm (the same coated mirror used for the standard FP cavity experiments). In the transmission maps, one can see an anti-crossing region forming

around the 1.43 THz vibrational resonance of glucose. Additionally, as a cavity mode shifts towards the MS/vibrational resonance, the linewidth of the cavity mode can be shown to narrow slightly. The bottom row of Fig. 4(b) compares the transmission characteristics of the H1 architecture when a cavity mode is overlapped with the MS/vibrational resonance (on-resonance, red curve) versus when a cavity mode is spectrally far from the MS/vibrational resonance (off-resonance, blue curve). We observe the formation of polariton peaks when the cavity mode is resonant with the MS/vibrational resonance. Since the glucose molecules within the H1 configuration are not within the plasmonic mode volume of the MS, the MS just acts as a frequency selective mirror. As a result, the cavity mode dominates the photonic response of the hybrid cavity, and the frequency splitting and FF is the same as observed in the standard FP cavity arrangement, which is 140 GHz and ~80%, respectively.

The H2 architecture, with results shown in Fig. 4(c), is particularly suited for the study of complex polaritonic systems as it couples the glucose resonance to both the cavity-delocalized mode and to the MS resonance. For this configuration, we incorporate the same coupled metasurface studied in Fig. 3 (coating layer (3)) into a hybrid cavity with an uncoated planar mirror. A coupled notch filter MS shows polaritons as minima in transmission, so resultant, we observe the same transmission characteristics in this hybrid configuration since the glucose is still within the plasmonic mode volume of the metasurface, dominating the coupling interaction. First, when the cavity mode is spectrally far from the bare resonance of glucose, we find the same polaritonic splitting as observed with the MS/glucose architecture in Fig. 3. Then, as the cavity mode shifts towards the glucose resonance, the polaritonic splitting increases (SI, see section *Constituent transmission curves for H2*). Transmission spectra for the on- and off-resonant cases are plotted in the bottom row of Fig. 4(c). We can observe how in the on-resonant case, the response of the H2 architecture shows an enhancement of the MS-glucose interaction, which leads to an effectively larger Rabi splitting of 200 GHz for the MS-glucose polaritons. Furthermore, with only 90 μm of glucose in this configuration, we calculate a FF of ~85%. The effect of the cavity mode is to enhance the Rabi splitting due to the additional field enhancement it provides. We discuss the origin of this polaritonic signature in more details using numerical simulations in the subsequent section.

7. It is not clear whether the three architectures have the same filling fraction (i.e. ratio between the volume of the glucose and the total volume between the mirrors). I believe that comparing these filling fractions is essential when claiming that the interaction strength is enhanced.

R1-7: We thank the Reviewer for the suggestion. In this work, we always try to maximize the density of matter contained within the electromagnetic mode volume to obtain the largest coupling strength as described in our *Setup and approach* section. Given that the density of glucose deposited with spray coating is reproducible and can be considered the same in all structures, we have now investigated the filling fraction for all architecture designs presented in our work within a new section of the supplementary information. Furthermore, we modify the text in the manuscript to explicitly indicate the definition of filling fraction.

[P. 6] Since the coupling strength scales with the filling fraction (FF), $FF = \sqrt{N/V_{opt}} = \sqrt{\rho V/V_{opt}}$, the glucose layer must ideally fill up the plasmonic mode volume to ensure that the largest number of emitters are coupled with the electromagnetic mode.

[P. 6] The measured intensity transmission of the coated MS is plotted in Fig. 3(c). [...] For the design (3) in Fig. 3(c), we calculate a FF of ~99% due to the strong overlap of the glucose region with the evanescent field of the MS (see SI, section *Glucose filling fraction*).

[P. 7] For a standard FP cavity configuration, one would expect the field amplitude to decay completely at the planar mirror interface and one would suspect that the molecular layer should be placed at the anti-nodes of the intracavity field. For this experiment, we calculate a FF of ~80%. Given the overall effective cavity spacing, the cavity field supports multiple modes and thus several anti-node positions. Therefore, coating a relatively thick molecular layer on one of the cavity mirrors provides a simple design approach to reach the strong light-matter regime with this architecture.

[P. 9] Since the glucose molecules within the H1 configuration are not within the plasmonic mode volume of the MS, the MS just acts as a frequency selective mirror. As a result, the cavity mode dominates the photonic response of the hybrid cavity, and the frequency splitting and FF is the same as observed in the standard FP cavity arrangement, which is 140 GHz and ~80%, respectively.

[P. 9] [...] the response of the H2 architecture shows an enhancement of the MS-glucose interaction, [...] Furthermore, with only 90 μm of glucose in this configuration, we calculate a FF of ~85%. The effect of the

cavity mode is to enhance the Rabi splitting due to the additional field enhancement it provides. We discuss the origin of this polaritonic signature in more details using numerical simulations in the subsequent section.

New Supplemental Information section: *Glucose filling fraction*

As discussed in the main text, there are several parameters that determine the strength of a light-matter interaction. Namely, the coupling strength depends on the transition dipole moment, the orientation between the matter dipole and the photonic mode polarization, the zero-point electric field amplitude, the square of the number of atoms/molecules, and the spatial distribution of the photonic mode. Furthermore, the zero-point electric field amplitude depends on the square of frequency and the inverse square of photonic mode volume. In our experiment, we target a vibrational mode of glucose for light-matter interaction. The transition dipole moment and the frequency of the molecular vibration are fixed parameters. Our polarization-insensitive MS/mirror design and orientation independent spray coating deposition technique allows us to neglect the dependence on orientation between the matter dipole and photonic mode polarization. Additionally, the photonic mode volume is fixed upon choosing a platform (MS, FP cavity, hybrid cavity). The only controllable parameter we are left with is the number of glucose molecules within the mode volume of our photonic platform. Our spray coating technique presents an opportunity to fine-tune the density of glucose which can be coated within the mode volume of our various architectures. In this work, the MS and planar mirrors are coated with the same spray coating procedure and glucose batch, so the density should be relatively consistent across the experiments. Optical microscope images of the glucose coatings on the MS and planar mirror samples are shown in Fig. S7. The regions of the glucose coating are indicated with red dashed lines and the regions of the MS/planar mirror are indicated with blue dashed lines for clarity.

Fig. S7: Optical microscope images of the coating optical components. (a) Image of a cross resonator array MS on a Zeonor substrate ($\approx 188 \mu\text{m}$, red dashed region) which is spray coated with a glucose coating (blue dashed region) that we estimate to be $\approx 90\mu\text{m}$. This is the MS that is utilized in the experiments of the paper. (b) Microscope image of a gold coated (9nm) GaAs substrate ($\approx 650 \mu\text{m}$, red dashed region) which is spray coated with a glucose coating (blue dashed region) that we estimate to be $\approx 210 \mu\text{m}$. This forms the coated mirror sample utilized in the experiments of the paper.

Another method of comparison between the various architectures can be to monitor the filling fraction (FF) of glucose within the photonic mode volume. Here, as stated in the main text, we consider the FF to be

$$FF = \sqrt{\frac{\rho V}{V_{opt}}}, \quad (\text{S.21})$$

where V_{opt} is the electromagnetic mode volume and ρ and V are the density and volume of glucose. Since our spray coating technique is quite uniform in area and thickness (with respect to the scale of THz wavelengths), we can consider the filling fraction to be proportional to the ratio of glucose thickness and the extent of the photonic mode in space (distance). We express the FF to be a ratio of field integrals over the region filled by glucose to the total region spanned by the electromagnetic mode volume

$$FF \approx \frac{\int |E_{gl}| dz}{\int |E_{opt}| dz}, \quad (\text{S.22})$$

where $|E_{gl}|$ represents the field magnitude within a 1D column that spans the region in the architecture which contains glucose and $|E_{opt}|$ spans the total electromagnetic region which interacts with glucose. We calculated the FF with numerical integration following FDTD simulations on the various architectures which are plotted in Fig. S8. The glucose thickness and air spacing used for the simulations match what we measured/calculated experimentally. The glucose index is taken to be frequency independent with an index of $n_{gl} = 1.9$.

Fig. S8: Plots of 1D field magnitude simulations of the FP, MS, and H2 architectures containing glucose. The position of the 1D columns for the structures containing a MS is chosen at a position which is representative of the overall field contained within an xy plane above the MS. The total field across the electromagnetic mode volume is indicated by the red curve and the portion of the field within glucose is indicated by blue dashed lines and a filled blue area beneath the curve. The filling fraction for each architecture is calculated using (S.22).

The FP cavity and H1 architectures have the largest volumes of glucose ($210 \mu\text{m}$) but will have the smallest FF because the electromagnetic field which interacts with glucose is not strongly confined to just the region of glucose. To reiterate, for H1, the MS effectively does not couple to glucose at all and just acts as a mirror which is why this architecture is equivalent to the FP cavity for the same glucose volume and density. The MS and H2 architectures have the strongest FFs with $90 \mu\text{m}$ of glucose. While the FF of H2 is lesser than the MS, the additional field enhancement within the glucose filled region of H2 allows for a substantially stronger light-matter interaction strength (200 GHz vs. 90 GHz) which we see in the experiment/calculation.

Subsection A: *Overfilling a metasurface*

For the MS in particular, we found it crucial not to overfill the resonator mode volume with glucose molecules to avoid a signal dominated by absorption. Below, we plot transmission spectra of a resonator as it is overfilled with glucose:

Fig. S9: Transmission spectra of a spray coated MS in which the mode volume is overfilled. A MS with a resonance frequency of 1.66 THz (black curve) is covered by successive layers of glucose while we observe a gradual redshift of its resonance frequency s . Once the plasmonic mode volume is completely filled with glucose we observe no more redshifting and the light-matter interaction cannot be further increased. Further coating the MS only causes the transmission to become absorption dominated (magenta curve).

In the main text, the glucose-coated MS has an initial resonance frequency of 1.6 THz. While filling the mode volume with glucose, the plasmonic resonance redshifts to 1.43 THz, the vibrational resonance of glucose and couples strongly. In the plot above, we show the results obtain using a MS with a bare resonance frequency of 1.66 THz. As we deposit successive glucose layers on this MS, we can also redshift the resonance frequency to 1.43 THz. However, more glucose is required and the linear absorption spectrum of glucose starts to dominate the transmission spectrum, which is detrimental to monitoring the polaritonic signature.

Reviewer 2:

In their manuscript titled “Hybrid architectures for molecular polaritonics” by Jean-Michel Ménard et al, the authors report fabrication, simulation and experimental characterization of what they call hybrid architectures to enhance light-matter coupling between dipole resonances in complex molecules and light at terahertz (THz) frequencies.

I like this work, but in many respects, it needs to be substantially improved to meet the publication standards set forth by Nature family of journals. With sufficient reworking of the manuscript, I hope the authors can improve the quality to an adequate level. I have for you below an itemized summary of my feedback that guides my decision.

R2-0: We appreciate that the Reviewer’s constructive criticism and we think that the current draft, greatly improved owing to the comments from the Reviewers, is now a better fit to Nature Communications.

- The novelty of this work needs to be adequately motivated. It is not clear to a non-specialist in this field whether the metasurface approach used for mirrors in (molecular) polaritonics has been tried in prior works
- it seems like a straightforward idea, especially at low frequencies, where metasurface fabrication tolerances are pretty relaxed. If this was explored by others, I failed to see a discussion dedicated to the crystallization of the novelty of this work in the context of the prior reports.

R2-1: We thank the Reviewer for this comment. The novelty aspects of our work are now explicitly stated in the *Introduction* and *The nature of the hybrid cavity enhancement* sections of the manuscript. We first do so by comparing our work to general experiments reporting on light-matter coupling of THz vibrational resonances and other work exploring hybrid cavity architectures. We then emphasize that our unprecedented design, using a reflective metasurface to replace one of the mirrors of a standard planar cavity, enables a new electromagnetic mode volume enhancing light-matter interaction when molecules are in direct contact with the plasmonic metasurface (see also R1-0, R1-1 and R1-2). Finally, we believe that this innovative hybrid cavity design will spark new studies and applications leveraging molecular polaritonics at all optical frequencies, but especially in the THz region where high-quality MS can be easily fabricated.

[P. 1] Within regions of lower energies within the electromagnetic spectrum, light can also directly couple to intra- and inter-molecular vibrational modes¹⁰⁻¹². This can lead to emerging applications in chemistry through the modification of ground state potential landscapes allowing for the manipulation of chemical reaction pathways and rates¹³⁻¹⁵. However, for the purpose of studying light-matter coupling involving far-infrared light and low-energy vibrational transitions of molecules, the DBR design becomes unpractical since most dielectrics are absorptive in this region and their required thickness exceeds the capacity of most nanofabrication equipment. In the THz region (wavelengths from 0.1 to 1 mm), the coupling to low energy bonds, such as intermolecular vibrations or hydrogen bonding, becomes accessible^{16,17}. At these larger wavelengths, Fabry-Perot (FP) cavities utilizing metallic planar mirrors have been demonstrated, where a VRS on the order of 68 GHz has been reached¹⁸. Following this framework, more sophisticated cavity architectures can potentially lead to even stronger polaritonic splitting with molecular systems in the THz region.

In the context of light-matter interaction with solid-state emitters, different types of THz resonators based on plasmonic emitters/antennas^{19,20}, waveguides^{21,22}, and metasurfaces (MSs)²³⁻²⁵, or a combination of them²⁶⁻²⁹, have been explored to achieve a strong field confinement. However, THz cavity architectures involving multiple electromagnetic resonators have not yet been explored in combination with resonant molecular ensembles, although few of these structures optimized for the mid-infrared range have been investigated^{30,31}. Thus, a systematic study of metasurface-based cavities in the emerging context of THz molecular polaritonics is needed to enable a range of chemistry applications such as those discussed above.

Here we propose a set of four THz resonator architectures illustrated in Fig. 1, including two hybrid designs relying on a reflective MS (analogous e.g., to subradiant optical mirrors realized with trapped atoms³²) replacing the planar mirror of a FP cavity to achieve better cavity finesse³³. These designs are used to yield unprecedented electromagnetic field spatial distributions peaking at the interface of the MS. We explore the coupling of these fields to a resonant transition of glucose, a molecule with a strong relevance to biological processes including metabolism and photosynthesis. Until now, the coupling of molecules with hybrid cavities has remained mostly unexplored, in part due to the complexity of fabricating a dense layer of organic material within a spatially confined electromagnetic mode volume. Using a spray coating technique³⁴, we obtain a dense molecular layer in direct contact with the surface of the MS allowing us to reach the strong light-matter coupling regime involving a resonant plasmonic and molecular vibrational mode. This system can then be inserted in the design of a planar resonator geometry to enable further hybridization with a photonic mode.

We experimentally and numerically compare the transmission properties of such cavity architectures, exploring the evolution from a standalone resonator towards a hybrid design with and without (see Supplementary Information (SI), section *Empty hybrid cavity*) the presence of a resonant molecular transition. This systematic experimental study supported by theory and simulations, provides a unique window into the complex interaction between plasmonic, photonic and vibrational resonances. Most notably, we compare the polaritonic splitting achieved with a plasmonic MS and a standard FP cavity design. Interestingly, a similar Rabi splitting is observed although fewer molecular emitters can be contained within the evanescent mode volume of the MS, in comparison to those within the FP cavity mode. This indicates that the metasurface, through a larger mode confinement and stronger field enhancement, enables an enhanced coupling per molecule. Then, using a hybrid cavity architecture, where the MS replaces one of the mirrors of the FP cavity, we reach a 40% larger polaritonic splitting with less glucose volume than a standard FP due to a modified intracavity field.

As the Reviewer mentions, at lower frequencies, metasurface fabrication tolerances are relaxed which makes them attractive platforms to be used as photonic resonators. Several experiments show THz metasurfaces coupled with artificial matter resonances (quantum wells, dots, 2DEG, etc...) and we have added these references to our manuscript. However, we note that the exploration of light-matter coupling with organic

molecules in the THz has been limited to a standard Fabry-Perot cavity configuration (R. Damari, et al., Nat Commun., 10, 3248 (2019)). In our experience, the biggest challenge is not in fabricating the THz metasurface, but rather in depositing a sufficient high density of organic molecules at the interface of the resonance to allow sufficient coupling strengths. We used a spray coating technique in this work, which perfectly fulfill this condition. We now explicitly state the benefit of our spray coating technique to molecular polaritonics in the manuscript. This is a novel fabrication technique in this context, and we believe that it can be broadly applicable for creating films of crystalline molecules to investigate a range of strong coupling experiments at THz frequencies.

We add these details to the section *Coupling with a metasurface resonator*:

[P. 6] We make use of a spray coating technique, which allows us to deposit thin films of crystalline molecules with subwavelength grain structures directly within the plasmonic mode volume of our metasurfaces (see SI, section *Sample fabrication* for procedure outline) The challenge of studying organic molecular polaritonics with a THz MS is not the fabrication of the metasurface, but rather in depositing a sufficiently high density of organic molecules at the interface of the MS to allow sufficient coupling strengths; our technique fulfills this condition while being robust.

The authors remark that THz range is less explored, but then the paper's focus should be narrowed down to that range, including already in the title (which is, at present, too broad and hence is in effect over-claiming the anticipated impact).

R2-2: Indeed, our work investigate the regime of strong light-matter coupling in the THz region, which has been far less explored than IR or visible regions, as acknowledged by the Reviewer. As such we have added “terahertz” in the title, which now reads: **Hybrid architectures for terahertz molecular polaritonics**, to indicate this novelty aspect of our work. However, our work also presents new ideas that can be directly applied to the regime of strong light-matter coupling in the visible, near-infrared or mid-infrared regions. Notably, our work reports, for the first time, a hybrid cavity design using a metasurface as one of the mirrors of a planar cavity to exploit a new electromagnetic field distribution and achieve stronger light-matter interactions (as theoretically introduced and discussed by one of the authors, albeit in the optical regime and in the context of cavity quantum electrodynamics [Phys. Rev. Lett. 122, 243601 (2019)]. We now discuss explicitly this novelty aspect in the text (see also R1-0 and R2-1):

[Abstract] The terahertz hybrid cavity architecture designs presented in this work can be applied broadly to the visible, near-infrared, or mid-infrared regions of the electromagnetic spectrum to explore enhanced molecular polaritonic interactions.

[P. 2] In the context of light-matter interaction with solid-state emitters, different types of THz resonators based on plasmonic emitters/antennas^{19,20}, waveguides^{21,22}, and metasurfaces (MSs)²³⁻²⁵, or a combination of them²⁶⁻²⁹, have been explored to achieve a strong field confinement. However, THz cavity architectures involving multiple electromagnetic resonators have not yet been explored in combination with resonant molecular ensembles, although few of these structures optimized for the mid-infrared range have been investigated^{30,31}. Thus, a systematic study of metasurface-based cavities in the emerging context of THz molecular polaritonics is needed to enable a range of chemistry applications such as those discussed above.

Here we propose a set of four THz resonator architectures illustrated in Fig. 1, including two hybrid designs relying on a reflective MS (analogous e.g., to subradiant optical mirrors realized with trapped atoms³²) replacing the planar mirror of a FP cavity to achieve better cavity finesse³³. These designs are used to yield unprecedented electromagnetic field spatial distributions peaking at the interface of the MS. We explore the coupling of these fields to a resonant transition of glucose, a molecule with a strong relevance to biological processes including metabolism and photosynthesis. Until now, the coupling of molecules with hybrid cavities has remained mostly unexplored, in part due to the complexity of fabricating a dense layer of organic material within a spatially confined electromagnetic mode volume. Using a spray coating technique³⁴, we obtain a dense molecular layer in direct contact with the surface of the MS allowing us to reach the strong light-matter coupling regime involving a resonant plasmonic and molecular vibrational mode. This system can then be inserted in the design of a planar resonator geometry to enable further hybridization with a photonic mode.

- One of the parameters of light-matter coupling should be the overlap of the photonic density of states with the position of the intracavity layer containing the matter part. This is not explained at all, however both the thickness of the matter layer and its position can drastically affect the overall coupling to a standing photonic

wave. A naïve expectation would be that the optical field magnitude at a position of a (perfect) mirror should be zero, hence placing a layer near the cavity boundaries would not be a textbook choice. Presented simulations, shown toward the end of the manuscript, seem to disprove this picture, however no discussion is given. The effect is likely due to the complex response function of the MS and, in fact, tunability of these effects could have been elucidated as one of the advanced approaches for one would use MS in the “hybrid” designs.

R2-3: In the case of a Fabry-Perot cavity, constructed from planar mirrors, the “textbook” choice would indeed be to place the matter layer at the antinodes of the optical field within the cavity as the reviewer correctly states. In our experiment, the cavity spacing is large enough to support higher order modes with multiple antinodes and the molecular layer nearly fills 80% of the cavity volume (see also R1-6). We now included this information in the manuscript. In the case of a hybrid cavity, simulations shown in Fig.5(b) demonstrate how the hybrid cavity H2 does not have a similar intracavity field response to that of a standard Fabry-Perot cavity. Unlike a mirror, the metasurface has its maximum field amplitude just above its surface. In this case, depositing the glucose layer directly on the MS is a simple and efficient fabrication technique to fill up the plasmonic mode volume and achieve the maximum coupling strength. We also concur with the Reviewer that the metasurface design adds a tunability knob to the hybrid cavity design which we now emphasize explicitly in the section *Coupling with a standard Fabry-Perot cavity*:

[P. 7] For a standard FP cavity configuration, one would expect the field amplitude to decay completely at the planar mirror interface and one would suspect that the molecular layer should be placed at the anti-nodes of the intracavity field. For this experiment, we estimate a glucose *FF* of ~80%. Given the overall effective cavity spacing, the cavity field supports multiple modes and thus several anti-node positions. Therefore, coating a relatively thick molecular layer on one of the cavity mirrors provides a simple design approach to reach the strong light-matter regime with this architecture.

[P. 8] The plasmonic field of the MS peaks at the interface and decays exponentially with the distance (see SI, section *Response of a plasmonic metasurface*). Therefore, a strong light-matter regime can be achieved when molecules, with resonant transitions, are deposited directly on the MS. In a hybrid cavity architecture using a MS as a reflective element, the field distribution also has a similar evanescent field distribution at that interface, which is demonstrated with numerical simulations in the subsequent section. Experimentally, we demonstrate that we can reach the strong light-matter coupling regime by filling up this plasmonic field volume with glucose in a hybrid cavity, to obtain an even larger polaritonic splitting frequency than the one observed with a bare MS or FP resonator design. We also compare this hybrid cavity architecture to a similar one in which the glucose layer is deposited on the surface of the planar mirror instead of the MS reflector. Additionally, we compare this hybrid cavity architecture to a similar one in which the glucose layer is deposited on the surface of the planar mirror instead of the MS reflector and also the empty hybrid cavity without any glucose layer (see SI, section *Empty hybrid cavity*).

- THz measurements here are reported as “time-resolved”, however this can be misleading to a non-specialist audience. Instead, EOS here is used instead of an FTIR or any other standard spectroscopic analysis. It is not clear why the EOS measurement is used here, is it due to better SNR compared to say FTIR? What is the frequency range over which EOS is operating?

R2-3a: We thank the Reviewer for raising this point. A time-resolved THz spectroscopy system relying on EOS is used here, instead of a THz-FTIR system, to achieve the best signal to noise ratio around 1.5 THz. The sensitivity of FTIR systems in this range is low. We have added a sentence to justify our choice of characterization technique.

[P.4] For our investigation of the vibrational resonances of glucose interacting with THz resonators, we focus our measurement range to the low THz frequency range between 0.5 and 3 THz. In this range, we find that EOS is a more suitable detection technique than the standard Fourier-transform infrared (FTIR) spectroscopy, which is broadly used for sample characterization in the mid-IR. We found that EOS offers a larger signal-to-noise ratio than THz-FTIR systems we tested around the bare resonance frequency of 1.43 THz in our experiments.

Since the technique measures both real and imaginary components of the electric field, why is the phase response (dispersion of the refractive index) of the sample not shown in the main text?

R2-3b: The phase response of the metasurface and hybrid cavities could not be extracted with sufficient accuracy in our experiments. These structures are relatively thin, e.g. the metasurface is 200 nm-thick, which prevents a significant time delay to be extracted to retrieve the real part of the refractive index and, consequently, the dispersion. Note however, that we could measure both real and imaginary part of the refractive index of a 300 μm -thick glucose pellet as shown in Fig. S1 and discussed in section *Sample fabrication* of the SM.

Is Fig1 of the supplementary showing this? If it is (not clear from the current narrative) – make sure you properly reference figures/results in the Supplementary section in the main text or better yet transfer this to the main part and draw more emphasis to this, effectively justifying EOS also. (Aside: n_g notation is typically reserved for the group refractive index, so it might be confusing when n_g is quoted in the text in a different context. Perhaps the ambiguity can be omitted with “phase refractive index of glucose (n_{gl}) is given by ...”.)

R2-3c: Fig. 1a of the supplementary information shows the real and imaginary parts of the refractive index of glucose. We utilized the imaginary index to calculate the absorption spectra of glucose, which is presented in the inset of Fig. 2. We have clarified the relationship explicitly in the manuscript with a reference to the Supplemental Information. Also following the Reviewer’s suggestion, we now use n_{gl} in the text to refer to the refractive index of glucose, instead of n_g .

[P.4] The inset of Fig. 2 shows the absorption spectrum of glucose with a prominent vibrational resonance at 1.43 THz and background absorption increasing towards higher frequencies. We utilize phase and amplitude information provided by the EOS detection technique to directly extract the real and imaginary parts of the refractive index of glucose (see SI, section *Sample fabrication*). The real part of the refractive index of glucose is $n_{gl} = 1.9$ at 1.43 THz.

- My understanding of the critical aspect of the hybrid design is the inclusion of a resonant effect in the frequency response of the metasurface (even though I fail to see a clear justification for this). But my main concern for the presentation is why t’e MS’s resonance hasn’t been measured in the same manner as that of the glucose layer. It would have been very instructive to compare both responses of the “bare” cavity, signal that contains only MS resonances, with that of the “loaded” cavity, now containing the anti-crossing behavior in the hybridized modes. Some version of that is shown in Fig 2, but again, the response of the MS (important for this discussion)’hasn’t been properly isolated.

R2-4: To address the first part of the Reviewer’s point, we now explicitly mention in the text that the glucose absorption and MS transmission spectra were all measured with time-resolved THz spectroscopy.

We believe the second part of the Reviewer’s comment concerns the hybrid cavity design containing a MS. We collected experimental measurements and have added an entirely new section in the supplementary information (*Empty hybrid cavity*) that compares the results obtained with H1 and H2, referred by the Reviewer as “loaded” cavities, with the bare hybrid cavity containing no glucose. Numerical calculations comparing a bare hybrid cavity with a Fabry-Perot cavity are also shown in supplementary section *Transfer matrix theory*. We also refer to this study in the main text within the *Coupling with hybrid cavity architectures* section:

[P. 6] The measured intensity transmission of the coated MS is plotted Fig. 3(c). The transmission spectra of these measurements are obtained via time-resolved THz spectroscopy.

[P. 7] While strong coupling can be achieved in both setups described above, integrating MSs with flat mirrors can bring a few advantages. One stems from the convolution of the MS’s narrow frequency response with the cavity transmission window to design sharper resonances³³ and thus sharper polaritonic peaks. Another arises from the interaction of three resonances involving the glucose, MS and cavity, to yield a potentially richer polaritonic physics beyond the standard upper and lower polariton states typical of strong coupling experiments. An additional advantage is the ease in cavity design complexity allowing one to directly coat one of the cavity mirrors with a molecular/atomic ensemble rather than position a metasurface within a cavity field³⁸; this advantage arises from the unique mode distribution and light-matter interaction enhancement which we explore in the subsequent section.

[P. 9] Additionally, we compare this hybrid cavity architecture to a similar one in which the glucose layer is

deposited on the surface of the planar mirror instead of the MS reflector and also the empty hybrid cavity without any glucose layer (see SI, section *Empty hybrid cavity*).

New Supplemental Information section: *Empty hybrid cavity*

To complement the experimental measurements on the H1 and H2 cavity architectures in the paper, we perform measurements of an empty hybrid cavity (no glucose) composed of a reflective metasurface facing a partially reflective metallic mirror. We discuss and plot the numerically calculated field distribution within such an empty hybrid cavity in Fig. 5(b) of the paper. In Fig. S3, we plot the transmission response of an empty hybrid cavity in comparison to the response of a Fabry-Perot cavity. Notably, the resonance of a hybrid cavity experiences a frequency shift and displays a Fano-like spectra. Fig. S6 shows three plots of experimental empty hybrid cavity measurements. In Fig. S6(a), we show the transmission of a standalone metasurface with a resonance frequency of 1.45 THz and the transmission of a planar mirror used in our hybrid cavity experiments. A vertical grey line is plotted along the resonance frequency position of the metasurface. In Fig. S6(b), the hybrid cavity is formed and we show a transmission map of the experimental measurements. Around the overlap between the cavity mode and shifted metasurface resonance mode, we observe an anti-crossing feature. Prior works on empty hybrid cavity designs also demonstrate splitting just by the nature of having a plasmonic mode interacting with a cavity mode [D. G. Baranov et al., Nat Commun., 11, 2715 (2020)]. In Fig. S6(c), we plot a few cavity spacings when the MS resonance is brought into overlap with the cavity mode resonance (green to red). We keep the vertical grey line showing the initial resonance frequency of the standalone MS. The cavity mode forms because the metasurface is acting as a frequency selective mirror. Interestingly, it seems that the resonance frequency of the MS shifts from its initial frequency and couples with the cavity mode to produce hybrid states.

Fig. S6: Experimental measurements of the empty hybrid cavity. (a) comparison of the response of a planar mirror, created from sputtering gold on GaAs, and a notch resonator metasurface array with a resonance around that of glucose. (b) An empty hybrid cavity is constructed from the metasurface and planar mirror. A transmission map

shows an anti-crossing feature around the overlap between a cavity mode and the metasurface resonance frequency.
(c) A few measurements of varying relative cavity spacings are shown as a cavity mode is brought into spectral overlap with the metasurface mode.

- In the presented form of the Hamiltonian, the importance of the angle formed between the polarization of light and the orientation of the dipole is briefly mentioned. However, for a complete and balanced presentation, one needs to include a discussion on whether a specific orientation of the dipole moment of the glucose layer has been ensured (specifics of fabrication and/or post-processing?) and verified. This is an important step. Suppose the glucose response is isotropic: Why couldn't the effect be demonstrated more clearly by switching the polarization of light (using the 90deg rotation symmetry of the MS) from aligned to perpendicular orientation with the dipole?

R2-5: As a result of the sample fabrication technique relying on spray coating deposition, glucose molecules do not have a preferential orientation in our experiments. We now clarify this point in the manuscript:

[P. 5] The unit vectors $\hat{\epsilon}_\mu^j$ and $\hat{\epsilon}_c$ account for the relative orientation between the molecular dipole and the cavity polarization, respectively. Glucose molecules do not have a preferential orientation in our experiments since they are deposited by a spray coating technique. Furthermore, the metasurface design utilized as standalone and in the hybrid cavities is created to have 4-fold symmetry which allows its transmission spectrum to be independent of the incident THz polarization state. As a result, changing the orientation of the incident polarization relative to the metasurface does not have any impact on the results.

- Simulation results have been mentioned several times in the main text, however, it is surprising that the main experimental finding reported in Figure 2 (where the 200 GHz splitting is presented) is not shown alongside the numerics. It would strongly add to the clarity and credibility of the findings. In particular, the appearance of the 3rd peak in the main result showing 200 GHz anticrossing is not discussed in much detail. It is indeed an interesting feature and I would have liked to see a creative/insightful discussion about how can this feature (due to MS, correct?) be used to improve/advance/enhance understanding/applications of polaritonics.

R2-6: Here we assume that the Reviewer is referring to Fig. 4c, where the 200 GHz splitting is presented, instead of Fig. 2, which contains an inset showing the absorption spectrum of bare glucose.

The Reviewer makes a good point. We have now moved the numerical calculations/simulations corresponding to the results from the supplementary information to Fig. 4 of the main text. We also now show both experimental and simulation results of the MS transmission spectra, with and without glucose layer on top, in Fig. 3.

Reviewer 3:

The authors demonstrate strong coupling regime between THz vibration transition of glucose (1.5 THz) and a hybrid photonic architecture, which consists of a Fabry-Perot cavity where one of the mirrors is replaced with a meta-surface. The authors compare structures which consist of meta-surface alone, Fabry-Perot cavity alone, and a hybrid cavity, study the light-matter coupling constant which can be inferred from the optical spectroscopy of the system. Namely, the coupling strength is proportional to the spectral anti-crossing of the light-matter coupled (polariton) modes. The authors claim strong enhancement of the light-matter coupling constant in a hybrid cavity. As shown further, I strongly contest this result which seem to be a misinterpretation of the experimental data, as commented further.

R3-0: In the revised manuscript, we now clarify our interpretation of the results by adding a more detailed analysis and by highlighting the excellent agreement between our experimental measurements and calculations, e.g. see revised Fig. 4. With these changes, we believe we present the best physical interpretation of the experimental data. We also demonstrate below that the alternative scenario proposed by the Reviewer is not valid in our case.

Before detailing this point, I would like to point out that the phenomenon of strong coupling between THz vibrational transitions has been already been observed (<https://doi.org/10.1038/s41467-019-11130-y>).

R3-1: This study is extremely relevant to our work and has motivated us to expand the research on strong coupling with molecular vibrations in the THz regime with an improved configuration and analysis. It was for this reason that we explicitly cited this work in our manuscript (originally Ref. 16 and now Ref. 18). We now explicitly mention some of the key differences between this work and our work:

[P. 1] At these larger wavelengths, Fabry-Perot (FP) cavities utilizing metallic planar mirrors have been demonstrated, where a VRS on the order of 68 GHz has been reached¹⁸. Following this framework, more sophisticated cavity architectures can potentially lead to even stronger polaritonic splitting with molecular systems in the THz region.

Also, many THz hybrid photonic architectures have already been demonstrated (<https://doi.org/10.1021/acsphotonics.1c00717>, <https://doi.org/10.1364/PRJ.482195>, <https://doi.org/10.1364/OE.456044>). In particular, the last two references present a much more advanced analysis of hybrid structures than the one proposed by the authors (see Fig.5 and the corresponding discussion). Thus, there is a serious doubt that the paper contains sufficient novelty to grant a publication at sications.

R3-2: Our hybrid cavity, which is based on a reflective metasurface replacing the mirror of a standard planar metallic cavity, is fundamentally different from all of those previously reported because of the unique electromagnetic field spatial distribution that it can support. In addition, we demonstrate hybridization of an organic material resonance inside this cavity, as revealed by a large polaritonic splitting. Our analysis, supported by a systematic experimental analysis of the cavity configuration in addition to an analytical model and numerical simulations, is comparable to, or extends beyond, those presented in previous work.

Experiments on light-matter coupling with molecules in the THz is largely unexplored. One of the main reasons for the lack of studies in this area is the complexity of depositing a dense enough layer of molecules within a photonic mode volume. Another difficulty is in designing suitable photonic structures at THz wavelengths which are compatible with molecular deposition. In our work, we present a pathway towards solving both of these challenges which is unprecedented.

We now discuss more explicitly the novelty aspects of our cavity architectures relative to previous work on light-matter interaction using hybrid cavity designs and we emphasize that we investigate for the first time the use of metasurfaces and hybrid cavity architectures for the exploration of molecular polaritonic system. This is fundamentally different from the 3 papers mentioned by the Reviewer: The first one explores coupling between a modified Fabry-Perot cavity and a two-dimensional electron gas, the second explores the design of a THz Tamm cavity coupled to an LC metamaterial resonator, and the third paper explores coupling between a metamaterial and a photonic crystal cavity. We now cite these three papers in the context of hybrid cavity architectures to provide a more complete background picture. We also made major changes to the whole *Introduction* section, which now elaborate on the novelty aspects mentioned above and more clearly discuss the role of the hybrid cavity in shaping the electromagnetic field distribution towards achieving a stronger light-matter coupling. See also R1-0 and R2-1 for related discussion.

[P. 1] Within regions of lower energies within the electromagnetic spectrum, light can also directly couple to intra- and inter-molecular vibrational modes¹⁰⁻¹². This can lead to emerging applications in chemistry through the modification of ground state potential landscapes allowing for the manipulation of chemical reaction pathways and rates¹³⁻¹⁵. However, for the purpose of studying light-matter coupling involving far-infrared light and low-energy vibrational transitions of molecules, the DBR design becomes unpractical since most dielectrics are absorptive in this region and their required thickness exceeds the capacity of most nanofabrication equipment. In the THz region (wavelengths from 0.1 to 1 mm), the coupling to low energy bonds, such as intermolecular vibrations or hydrogen bonding, becomes accessible^{16,17}. At these larger wavelengths, Fabry-Perot (FP) cavities utilizing metallic planar mirrors have been demonstrated, where a VRS on the order of 68 GHz has been reached¹⁸. Following this framework, more sophisticated cavity architectures can potentially lead to even stronger polaritonic splitting with molecular systems in the THz region.

In the context of light-matter interaction with solid-state emitters, different types of THz resonators based on plasmonic emitters/antennas^{19,20}, waveguides^{21,22}, and metasurfaces (MSs)²³⁻²⁵, or a combination of them²⁶⁻²⁹,

have been explored to achieve a strong field confinement. However, THz cavity architectures involving multiple electromagnetic resonators have not yet been explored in combination with resonant molecular ensembles, although few of these structures optimized for the mid-infrared range have been investigated^{30,31}. Thus, a systematic study of metasurface-based cavities in the emerging context of THz molecular polaritonics is needed to enable a range of chemistry applications such as those discussed above.

Here we propose a set of four THz resonator architectures illustrated in Fig. 1, including two hybrid designs relying on a reflective MS (analogous e.g., to subradiant optical mirrors realized with trapped atoms³²) replacing the planar mirror of a FP cavity to achieve better cavity finesse³³. These designs are used to yield unprecedented electromagnetic field spatial distributions peaking at the interface of the MS. We explore the coupling of these fields to a resonant transition of glucose, a molecule with a strong relevance to biological processes including metabolism and photosynthesis. Until now, the coupling of molecules with hybrid cavities has remained mostly unexplored, in part due to the complexity of fabricating a dense layer of organic material within a spatially confined electromagnetic mode volume. Using a spray coating technique³⁴, we obtain a dense molecular layer in direct contact with the surface of the MS allowing us to reach the strong light-matter coupling regime involving a resonant plasmonic and molecular vibrational mode. This system can then be inserted in the design of a planar resonator geometry to enable further hybridization with a photonic mode.

We experimentally and numerically compare the transmission properties of such cavity architectures, exploring the evolution from a standalone resonator towards a hybrid design with and without (see Supplementary Information (SI), section *Empty hybrid cavity*) the presence of a resonant molecular transition. This systematic experimental study supported by theory and simulations, provides a unique window into the complex interaction between plasmonic, photonic and vibrational resonances. Most notably, we compare the polaritonic splitting achieved with a plasmonic MS and a standard FP cavity design. Interestingly, a similar Rabi splitting is observed although fewer molecular emitters can be contained within the evanescent mode volume of the MS, in comparison to those within the FP cavity mode. This indicates that the metasurface, through a larger mode confinement and stronger field enhancement, enables an enhanced coupling per molecule. Then, using a hybrid cavity architecture, where the MS replaces one of the mirrors of the FP cavity, we reach a 40% larger polaritonic splitting with less glucose volume than a standard FP due to a modified intracavity field.

Figure R1.

Next, I contest the author's interpretation of the polariton splitting. This is explained in the Figure above. In Figure (c) lower panel, there are actually 3 resonant features that can be attributed to polariton states (LP: lower polariton, UP: upper polariton). Comparing between the lower and middle panel of Figure (c), one can conclude that: a) the spectrum chosen by the authors does not actually correspond to the anti-crossing point; b) the UP is thus wrongly attributed to an adjacent photonic mode by the authors, and the UP polariton is actually the feature at much lower frequency as pointed out in red. Thus, the polariton splitting is much lower than the one claimed by the authors; actually a closer look at the middle panels in the Figure reveals that the polariton splitting is essentially the same in all three cases. Note that the analysis is rather complicated by the fact that the polariton anti-crossing is on the same order of magnitude (even lower) than the linewidth of the resonances. The best way to determine the polariton splitting would be to plot the positions of all observed resonances as a function of the cavity spacing that can be extracted from the middle panels. Such graph, combined a line indicating the 1.5 absorption of glucoses would allow to determine precisely the cavity spacing for which the anti-crossing appears.

R3-3: We disagree with the Reviewer's general interpretation of our data but understand that similarities between the middle graphs in Fig. 4 can lead to confusion. We agree that adding a line indicating the bare glucose absorption line can help identifying the polaritonic splitting and we have now added this line to Fig. 4. Doing so demonstrates that the upper polariton (UP) cannot be at the spectral position suggested by the Reviewer since it would then overlap almost perfectly with the bare resonance frequency of glucose. Vibro-polaritonic splitting should produce polaritons which are balanced around the resonance frequency of the matter and photonic modes. One can see this directly even from the abstraction to the Jaynes-Cummings model where the energy difference between upper and lower polaritonic states is given by (see, for example, R. F. Ribeiro et al., Chem. Sci., 9, 6325-6339 (2018)):

$$\hbar\Omega_{Rabi} = 2\hbar \sqrt{\frac{(\omega_{cavity} - \omega_{molecule})^2}{4} + g^2}$$

Here, Ω_{Rabi} is the Rabi splitting radial frequency, \hbar is the reduced Planck's constant, ω_{cavity} is the cavity

mode radial frequency, ω_{molecule} is the molecular vibrational mode radial frequency, and g is the system coupling strength. This equation simplifies to $\Omega_{\text{Rabi}} = 2g$ at zero detuning, requiring the polaritons to form balanced around the detuning frequency. Our analytical calculations (see supplement and the new manuscript Fig. 4) show that the transmissions curve which we plot at $\Delta d = 55 \mu\text{m}$ (red curve in Fig. 4, third column, fourth row) is roughly at the position of zero detuning so we would expect that the upper and lower polaritons should also be balanced around the vibrational resonance of glucose.

The figure below shows the positions of the LP and UP (purple arrows) as defined in our manuscript and the spectral position of the UP suggested by the Reviewer (brown arrow, labelled 'rUP').

Transmission of glucose on a mirror (black curve) and transmission of the H2 architecture (red curve) at zero detuning normalized in amplitude. The vertical dashed line indicates the bare resonance frequency of glucose. We can see here that the position attributed by the Reviewer to the UP corresponds almost exactly to the bare absorption line of glucose. The two symmetrical dips on each side of the bare resonance of glucose correspond to the upper and lower polaritons.

We also demonstrate that the Rabi splitting associated to these two dips scales with the coupling strength as shown in Fig. 5a. We now provide an extended discussion on the nature of the polaritons in the *Result section*, and we hope the Reviewer will agree that it supports and clarifies our original conclusions. See also changes in R1-6. Finally, following the Reviewer's request, we included in a new section of the SI: *Constituent transmission curves for H2* a waterfall plot corresponding to cross section of the color map in Fig. 4c.

New Supplemental Information section: *Constituent transmission curves for H2*

To visualize the evolution of the complex polaritonic response of the H2 cavity configuration, we plot the constituent transmission curves which were used to create the experimental transmission map of Fig. 4(c) as a waterfall plot in Fig. S5. Here, one can observe the evolution of the polariton dips from the coupled metasurface become spaced further apart due to a Rabi splitting enhancement as a cavity mode is brought into overlap.

Fig. S5: Waterfall plot of the transmission curves that make up the experimental transmission map of the H2 architecture shown in Fig. 4(c). The relative cavity spacing is gradually decreased from $\Delta d = 115 \mu\text{m}$ (bottom curve) to $\Delta d = 0 \mu\text{m}$ (top curve). As the photonic mode overlaps with the hybridized glucose-MS resonance, the Rabi splitting increases significantly. The black curve highlight the evolution of the polaritons, corresponding to transmission dips symmetrically positioned around the bare glucose resonance (grey dashed line).

Away from the photonic resonance (bottom curve), the transmission spectrum effectively shows a polaritonic splitting caused by the interaction of the glucose resonance at 1.43 THz (dashed line) with the MS plasmonic mode. This result agrees with the one shown in Fig. 3. As we decrease the cavity spacing and blueshift the photonic mode towards 1.43 THz, the upper polariton (UP) gradually shifts towards higher frequencies. A symmetrical behavior with the lower polariton (LP) shifting to lower frequencies is observed if we increase the cavity spacing so that the cavity mode redshifts towards the bare resonance frequency of 1.43 THz. These two dynamics are highlighted with thin black lines in Fig. S5.

We have also modified part of the results section in the manuscript surrounding the observations in Fig. 4 to clarify the nature of the polaritonic response. (See R1-6):

Coupling with a standard Fabry-Perot cavity. We systematically compare the performance of various cavity architectures shown in Fig. 4. The first row of Fig. 4 depicts, from left to right, the FP cavity filled with glucose, the MS/planar mirror configuration of H1 with glucose on the planar mirror, and the MS/planar mirror configuration of H2 with glucose on the metasurface. The second row of Fig. 4 shows transmission maps of these three cavity designs obtained with the transfer matrix method to visualize the anti-crossing behaviour indicative of the strong coupling regime. We use the relative change in cavity spacing, Δd , as a variable corresponding to the longitudinal displacement of one cavity mirror, which varies the air gap spacing between the two mirrors. The third row of Fig. 4 contains the corresponding experimentally extracted transmission maps showing good agreement with the model. The fourth row of Fig. 4 displays specifically selected transmission curves, including the zero-detuning cavity spacing, to highlight the polaritonic signatures.

Looking at the FP cavity architecture in Fig. 4 (a), one of the mirrors is coated with glucose using the aforementioned spray coating technique. THz-TDS measurements are taken at $5 \mu\text{m}$ increments of spacing between the cavity mirrors. This effective spacing is given by $d_{\text{eff}}(\omega) = d_{\text{air}} + d_{\text{gl}}n_{\text{gl}}(\omega)$, where d_{air} is the air space between the mirrors, $n_{\text{gl}}(\omega)$ is the refractive index of glucose, and d_{gl} is the thickness of the glucose layer. Analytically, one can deduce the expression for the transmission as

$$t_{FP}(\omega) = \frac{e^{-i\omega d_{eff}(\omega)/c}}{\zeta^2 + (1-i\zeta)^2 e^{-2i\omega d_{eff}(\omega)/c}} \quad (3)$$

where ζ describes the polarizability of the planar gold mirrors, which we assume to be frequency-independent. We find almost perfect agreement with the model and experimental measurements shown on the third row. The bright regions correspond to the cavity modes shifting in frequency with cavity spacing. An anti-crossing behavior is achieved around 1.43 THz, the bare resonance frequency of glucose, when one of the cavity modes shifts from 1.6 to 1.3 THz with increasing Δd . We observe the formation of two characteristic polaritonic peaks in transmission which are shown as two bright modes. A corresponding Rabi splitting of 140 GHz is observed. In the bottom row, we show the transmission of the mirror with a glucose coating (210 μm thick) and compare it to the transmission of the FP system at zero-detuning, which shows the symmetrical polaritonic splitting around the glucose transition. These results are consistent with prior work using a similar architecture with a pellet of lactose¹⁸.

For a standard FP cavity configuration, one would expect the field amplitude to decay completely at the planar mirror interface and one would suspect that the molecular layer should be placed at the anti-nodes of the intracavity field. For this experiment, we estimate a glucose FF of $\sim 80\%$. Given the overall effective cavity spacing, the cavity field supports multiple modes and thus several anti-node positions. Therefore, coating a relatively thick molecular layer on one of the cavity mirrors provides a simple design approach to reach the strong light-matter regime with this architecture.

Coupling with hybrid cavity architectures. While strong coupling can be achieved in both setups described above, integrating MSs with flat mirrors can bring a few advantages. One stems from the convolution of the MS's narrow frequency response with the cavity transmission window to design sharper resonances³³ and thus sharper polaritonic peaks. Another arises from the interaction of three resonances involving the glucose, MS and cavity, to yield a potentially richer polaritonic physics beyond the standard upper and lower polariton states typical of strong coupling experiments. An additional advantage is the ease in cavity design complexity allowing one to directly coat one of the cavity mirrors with a molecular/atomic ensemble rather than position a metasurface within a cavity field³⁸; this advantage arises from the unique mode distribution and light-matter interaction enhancement which we explore in the subsequent section.

The plasmonic field of the MS peaks at the interface and decays exponentially with the distance (see SI, section *Response of a plasmonic metasurface*). Therefore, a strong light-matter regime can be achieved when molecules, with resonant transitions, are deposited directly on the MS. In a hybrid cavity architecture using a MS as a reflective element, the field distribution also has a similar evanescent field distribution at that interface, which is demonstrated with numerical simulations in the subsequent section. Experimentally, we demonstrate that we can reach the strong light-matter coupling regime by filling up this plasmonic field volume with glucose in a hybrid cavity, to obtain an even larger polaritonic splitting frequency than the one observed with a bare MS or FP resonator design. We also compare this hybrid cavity architecture to a similar one in which the glucose layer is deposited on the surface of the planar mirror instead of the MS reflector. Additionally, we compare this hybrid cavity architecture to a similar one in which the glucose layer is deposited on the surface of the planar mirror instead of the MS reflector and also the empty hybrid cavity without any glucose layer (see SI, section *Empty hybrid cavity*).

The H1 architecture explored in Fig. 4(b) involves an uncoated MS with a resonance frequency of 1.45 THz merged with a glucose coated flat mirror with a coating thickness of 210 μm (the same coated mirror used for the standard FP cavity experiments). In the transmission maps, one can see an anti-crossing region forming around the 1.43 THz vibrational resonance of glucose. Additionally, as a cavity mode shifts towards the MS/vibrational resonance, the linewidth of the cavity mode can be shown to narrow slightly. The bottom row of Fig. 4(b) compares the transmission characteristics of the H1 architecture when a cavity mode is overlapped with the MS/vibrational resonance (on-resonance, red curve) versus when a cavity mode is spectrally far from the MS/vibrational resonance (off-resonance, blue curve). We observe the formation of polariton peaks when the cavity mode is resonant with the MS/vibrational resonance. Since the glucose molecules within the H1 configuration are not within the plasmonic mode volume of the MS, the MS just acts as a frequency selective mirror. As a result, the cavity mode dominates the photonic response of the hybrid cavity, and the frequency splitting and FF is the same as observed in the standard FP cavity arrangement, which is 140 GHz and $\sim 80\%$, respectively.

The H2 architecture, with results shown in Fig. 4(c), is particularly suited for the study of complex polaritonic

systems as it couples the glucose resonance to both the cavity-delocalized mode and to the MS resonance. For this configuration, we incorporate the same coupled metasurface studied in Fig. 3 (coating layer (3)) into a hybrid cavity with an uncoated planar mirror. A coupled notch filter MS shows polaritons as minima in transmission, so resultantly, we observe the same transmission characteristics in this hybrid configuration since the glucose is still within the plasmonic mode volume of the metasurface, dominating the coupling interaction. First, when the cavity mode is spectrally far from the bare resonance of glucose, we find the same polaritonic splitting as observed with the MS/glucose architecture in Fig. 3. Then, as the cavity mode shifts towards the glucose resonance, the polaritonic splitting increases (SI, see section *Constituent transmission curves for H2*). Transmission spectra for the on- and off-resonant cases are plotted in the bottom row of Fig. 4(c). We can observe how in the on-resonant case, the response of the H2 architecture shows an enhancement of the MS-glucose interaction, which leads to an effectively larger Rabi splitting of 200 GHz for the MS-glucose polaritons. Furthermore, with only 90 μm of glucose in this configuration, we calculate a FF of $\sim 85\%$. The effect of the cavity mode is to enhance the Rabi splitting due to the additional field enhancement it provides. We discuss the origin of this polaritonic signature in more details using numerical simulations in the subsequent section.

New Fig.4 which shows the resonance position of glucose for H1 and H2 for reader clarity and also includes the resultant transmission maps calculated through a transfer matrix approach.

New Fig.3 which includes transfer matrix calculation results along with the third glucose coating transmission measurements (red and orange dotted lines).

REVIEWER COMMENTS

Reviewer #1 (Remarks to the Author):

The authors have responded to all comments and revised their manuscript significantly. In particular, the discussion around the novel aspects of their results is much clearer now and I find the revised manuscript suitable for publication in Nature Communications.

Two minor additional comments:

1. In my previous comment #4 (on the sentence "one of the cavity modes..." I meant to ask which mode number.
2. In addition to the papers mentioned by Reviewer #3, a different architecture for THz strong coupling was demonstrated in <https://doi.org/10.1021/acsp Photonics.1c00309>. I believe this work is also quite relevant to the topic and should be cited by the authors where appropriate.

Reviewer #2 (Remarks to the Author):

The manuscript has now been sufficiently edited to improve the quality of its presentation, and I suggest publication of the manuscript.

The quality of the manuscript has improved considerably. The authors owe much to the tireless contribution of the reviewers, whose input ultimately made the work presentable. All newly presented arguments, key justifications and clarifications are now in place.

Reviewer #3 (Remarks to the Author):

The authors have definitely provided more details about their system. However, I still have very strong doubts about the claimed enhancement of the light-matter constant due to the meta-surface design. I do believe that the meta-surface has similar Rabi splitting to the other geometries, and even lower. The main difficulty in the interpretation of the data is the presence of many cavity modes, with a free spectral range on the order of the Rabi-splitting that the authors claim. The width of the resonances is also quite significant (although it must be acknowledged that here the glucose THz resonance has an improved quality factor with respect to previous works). The main mistake of the authors is that they consider a single mode light-matter coupling case and they force their interpretation within that picture, whereas a full understanding requires to consider the coupling of the single glucose resonance with the multiple equally spaced Fabry-Perot modes.

Actually, Figure S5 in the rebuttal letter is very helpful, as it clearly shows what is going on. In the bottom red curve, $Dd=115\mu\text{m}$, the glucose is in strong coupling with a FP mode 2 at 1.4THz. An FP mode 1 at about 1.1THz is visible in the spectrum. At 1.4 THz, the double dip structure comes from the two polariton states with a splitting of 100GHz that arise from the coupling between the FP mode 2 and the glucose. As the distance Dd is decreased down to $Dd=0\mu\text{m}$, the mode frequencies increase, and the FP mode 2 is shifted to higher frequencies. Now the FP mode 1 is shifted to 1.4THz where it is resonant with the glucose transition (top curve). Once again, one sees the two polariton modes that now arise from the coupling between the FP1 mode and the glucose with a slightly higher splitting, 110GHz. The splitting is higher because now the cavity has a slightly reduced mode volume.

What happens for the green curve? In that case the glucose transition is in between the FP mode 1 and the FP mode 2. There is a strong coupling with both modes, which yields three polariton states: lower polariton (LP), middle polariton (MP) and an upper polariton (UP). The LP is

essentially the FP mode 1, slightly shifted at lower frequencies, the UP is mode 2, slightly shifted at higher frequencies, and the MP is essentially the uncoupled glucose transition. But by no means can the difference between the UP and LP be used as an estimation of the coupling strength; this is true only for the case of a single mode coupled to a material transition. By the way, the authors can recover easily these type of spectra by considering a Hamiltonian (1) with two cavity modes (a_1, a_1+) and (a_2, a_2+) instead of one.

Of course, all experimental spectra can be recovered by the full electromagnetic classical model (equation (2) in the main text), which basically describes a dielectric slab with a resonant absorption placed in a multimode Fabry-Perot resonator. But this model do not correspond to the single cavity mode Hamiltonian (1).

This work this does not sustain its claims; there is very a serous misinterpretation of the experimental data. I do not recommend publication in Nature Comm.

We once again thank the Reviewers for their meticulous efforts to aid us in making substantial improvements to our manuscript pertaining to the context, readability, and presentation of data. We are very happy to see that Reviewers 1 and 2 are satisfied with our changes and we are eager to continue addressing any concerns that Reviewer 3 might express. We have addressed each Reviewer's remarks on a comment-by-comment basis, and we have made appropriate modifications to the text and figures of our manuscript.

In this document, the Reviewer's comments are in **black**, our responses are in **blue**, sections of the manuscript are in **purple**, and all changes to the manuscript are in **red**.

Reviewer 1:

The authors have responded to all comments and revised their manuscript significantly. In particular, the discussion around the novel aspects of their results is much clearer now and I find the revised manuscript suitable for publication in Nature Communications.

R1-0: We greatly appreciate the Reviewers assessment of our revised manuscript and their recommendation for publication in Nature Communications.

Two minor additional comments:

1. In my previous comment #4 (on the sentence "one of the cavity modes..." I meant to ask which mode number.

R1-1:

We now added this information.

[P. 7] An anti-crossing behavior is achieved around 1.43 THz, [...], when one of the cavity modes (of mode number $k = 4$) shifts from 1.6 to 1.3 THz

[P. 9] The H1 architecture explored in Fig. 4(b) involves [...] Additionally, as a cavity mode ($k = 4$) shifts towards the MS/vibrational resonance, the linewidth of the cavity mode can be shown to narrow slightly.

[P. 9] The H2 architecture, with results shown in Fig. 4(c), is particularly suited [...] Then, as the cavity mode ($k = 4$) shifts towards the glucose resonance, the polaritonic splitting increases (SI, see section *Constituent transmission curves for H2*).

2. In addition to the papers mentioned by Reviewer #3, a different architecture for THz strong coupling was demonstrated in <https://doi.org/10.1021/acsp Photonics.1c00309>. I believe this work is also quite relevant to the topic and should be cited by the authors where appropriate.

R1-2: We agree with the Reviewer on the relevance of the cited paper to our manuscript and have included it as a reference in the introduction:

[P. 2] In the context of light-matter interaction with solid-state emitters, different types of THz resonators based on plasmonic emitters/antennas^{19,20}, waveguides^{21,22}, **photonic crystal**²³, and metasurfaces (MSs)²⁴⁻²⁶, or a combination of them²⁷⁻³⁰, have been explored to achieve a strong field confinement.

Reviewer 2:

The manuscript has now been sufficiently edited to improve the quality of its presentation, and I suggest publication of the manuscript.

The quality of the manuscript has improved considerably. The authors owe much to the tireless contribution of the reviewers, whose input ultimately made the work presentable. All newly presented arguments, key justifications and clarifications are now in place.

R2-0: We are very pleased to hear that the Reviewer now suggests our revised manuscript for publication. We do appreciate the feedback that allowed us to significantly improve our manuscript. As such, we now

explicitly thank the Reviewers in our acknowledgement section.

“We thank the reviewers for their detailed and insightful comments on the manuscript.”

Reviewer 3:

The authors have definitely provided more details about their system. However, I still have very strong doubts about the claimed enhancement of the light-matter constant owe to the meta-surface design. I do believe that the meta-surface has similar Rabi splitting to the other geometries, and even lower.

The main difficulty in the interpretation of the data is the presence of many cavity modes, with a free spectral range on the order of the Rabi-splitting that the authors claim. The width of the resonances is also quite significant (although it must be acknowledged that here the glucose THz resonance has an improved quality factor with respect to previous works). The main mistake of the authors is that they consider a single mode light-matter coupling case and they force their interpretation within that picture, whereas a full understanding requires to consider the coupling of the single glucose resonance with the multiple equally spaced Fabry-Perot modes.

R3-0: We appreciate the acknowledgement of the Reviewer that we have provided more details regarding our system. We understand that the Reviewer is concerned about the potential interaction of multiple FP modes with the sugar absorption resonance. As mentioned by the Reviewer, the free spectral range ($FSR \cong 310$ GHz) of our cavity is on the order of the maximum polaritonic splitting observed in our experiment (200 GHz). Note that the spectral linewidth of any resonances considered in our work is < 70 GHz HWHM.

We believe that the Reviewer fails to consider that the FP modes, which we regard as not interacting with the sugar resonance, are both located 1 FSR away from resonance frequency. Therefore, the spectral separation between these neighbouring FP modes is $2*FSR \cong 620$ GHz. As schematically represented below at zero detuning and for the maximum polaritonic splitting of 200 GHz observed in our experiments, only the resonant FP mode (Mode 2 in figure below) interacts with the sugar absorption resonance since the neighbouring FP modes 1 and 3 are too far apart.

Actually, Figure S5 in the rebuttal letter is very helpful, as it clearly shows what is going on. In the bottom red curve, $Dd=115\mu\text{m}$, the glucose is in strong coupling with a FP mode 2 at 1.4THz. An FP mode 1 at about 1.1THz is visible in the spectrum. At 1.4 THz, the double dip structure comes from the two polariton states with a splitting of 100 GHz that arise from the coupling between the FP mode 2 and the glucose. As the distance d is decreased down to $d=0\mu\text{m}$, the mode frequencies increase, and the FP mode 2 is shifted to higher frequencies. Now the FP mode 1 is shifted to 1.4 THz where it is resonant with the glucose transition (top curve). Once again, one sees the two polariton modes that now arise from the coupling between the FP1 mode and the glucose with a slightly higher splitting, 110 GHz. The splitting is higher because now the

cavity has a slightly reduced mode volume.

What happens for the green curve? In that case the glucose transition is in between the FP mode 1 and the FP mode 2. There is a strong coupling with both modes, which yields three polariton states: lower polariton (LP), middle polariton (MP) and an upper polariton (UP). The LP is essentially the FP mode 1, slightly shifted at lower frequencies, the UP is mode 2, slightly shifted at higher frequencies, and the MP is essentially the uncoupled glucose transition. But by no means can the difference between the UP and LP be used as an estimation of the coupling strength; this is true only for the case of a single mode coupled to a material transition. By the way, the authors can recover easily these type of spectra by considering a Hamiltonian (1) with two cavity modes (a_1, a_1+) and (a_2, a_2+) instead of one.

R3-1: We copied below the figure referred to by the Reviewer, which consists of a hybrid cavity with one mirror of the planar FP cavity being replaced by a metasurface with a resonance at 1.43 THz, the glucose absorption line.

Contrarily to the Reviewer's claim, the red curve does not show any FP mode at 1.4 THz. We believe the Reviewer mistakenly attributed the position of a FP mode to a dip in transmission instead of the peak. One can directly consult Fig. 4(a) in the manuscript to see that in the case of a standard FP cavity design, the cavity modes appear as peaks, we state this in our discussion of the transmission map.

[P. 7] The bright regions correspond to the cavity modes shifting in frequency with cavity spacing. An anti-crossing behavior is achieved around 1.43 THz, the bare resonance frequency of glucose, when one of the cavity modes (of mode number $k = 4$) shifts from 1.6 to 1.3 THz with increasing Δd . We observe the formation of two characteristic polaritonic peaks in transmission which are shown as two bright modes. To further quell any future confusion regarding the standard FP cavity design, we have modified the text to reiterate the fact that photonic modes appear as peaks in transmission.

[P. 7] These results are consistent with prior work using a similar architecture with a pellet of lactose¹⁸, noting that both the photonic and polaritonic modes appear as peaks in transmission.

In this figure above, showing the HC2 architecture, the polaritonic splitting observed at 1.4 THz in the red curve is caused by a strong coupling between the glucose absorption line and the metasurface resonance as we discussed in the manuscript. This interaction has polaritons that appear as *minima* in transmission, unlike the case of a standard FP cavity.

[P. 9] A coupled notch filter MS shows polaritons as minima in transmission, so consequently, we observe the same transmission characteristics in this hybrid configuration since the glucose is still within the plasmonic mode volume of the metasurface, dominating the coupling interaction.

Finally, we explicitly mention in the surrounding discussion of Fig. 5(a) and Fig. 5(b) in the manuscript how we can model the H2 configuration as providing enhancement to the metasurface-glucose polariton splitting through both transfer matrix and finite-difference time-domain methods. Particularly in Fig. 5(b) we show that even an empty hybrid cavity configuration leads to a field enhancement close to the metasurface, which is one of the main conclusions of our experiments.

[P. 9] [...] When increasing the MS-glucose coupling, which is the case for H2 architecture, two dips appear, corresponding to the MS-glucose polaritons. Most importantly, the exhibited splitting is now roughly a factor of two larger than the case of coupling only to the MS. This enhancement can be traced back to the fact that the hybrid architecture leads to an increase in the zero-point electric field amplitude of the intra-cavity mode around the position where the glucose is added. To prove this point, we incorporate FDTD simulations [...]

The remainder of the Reviewer's discussion is also based on this incorrect assumption that we hope to have now clarified.

Of course, all experimental spectra can be recovered by the full electromagnetic classical model (equation (2) in the main text), which basically describes a dielectric slab with a resonant absorption placed in a multimode Fabry-Perot resonator. But this model do not correspond to the single cavity mode Hamiltonian (1).

This work this does not sustain its claims; there is very a serous misinterpretation of the experimental data. I do not recommend publication in Nature Comm.

R3-2: The standard understanding of strong coupling and the Purcell effect within the formalism of cavity quantum electrodynamics is based on a single mode Rabi or Jaynes-Cummings (after rotating wave approximation is made) Hamiltonians, for single particles and Dicke or Tavis-Cummings Hamiltonians, for many particles. While the Reviewer is correct in estimating that most cavities are not single mode, when the free spectral range is large enough, a single mode approach suffices. Moreover, a single mode approach is easily justified by comparison with the transfer matrix approach which we perform here.

If the Referee argues that the effects can be reproduced by classical calculations it is indeed true in this case: the strong coupling regime in the weak excitation is indeed a linear response theory effect. However, the formalism then allows to reveal quantum effects such as for example the photon blockade effect and aspects such as light-matter entanglement are well reproduced within the simple Tavis-Cummings Hamiltonian.

There are many tutorials showing how one can easily move from purely quantum formulations to standard transfer matrix models, like for example, N. Német et al., Phys. Rev. Applied, 13, 064010 (2020) or M. Reitz et al, PRX Quantum 3, 010201 (2022).

REVIEWER COMMENTS

Reviewer #3 (Remarks to the Author):

The main difficulty with the paper is the presentation of the results, which still persist in Figure 4 of the paper. The problem is the complexity of the structure, which presents many different modes that appear either as peaks (FP modes) or dips (MS modes), if I currently understand correctly their description. The light-matter coupling constant, as defined by the authors through the Hamiltonian (1), can be reliably extracted from spectroscopic data only in the case where there is a single mode interacting with the matter excitation, and all other modes of the system are sufficiently far so not to perturb this simple picture (see for instance <https://journals.aps.org/prl/abstract/10.1103/PhysRevLett.105.196402>, or <https://journals.aps.org/prb/abstract/10.1103/PhysRevB.86.125314>).

My main concern, which have not been addressed yet, is the proper estimation of the light-matter coupling constant in the case of the hybrid MS-FP structure. The Hamiltonian (1) corresponds to the case of a single cavity resonance coupled with the glucose transition. In that case, there is a single coupling parameter g_{eff} that can be extracted from the data and compared to other system.

However, in the case of FP-MS resonator the situation seems to be more complex, and the model (1) does not apply any more. The authors acknowledge that fact: they state in the main text “[strong coupling]... Another arises from the interaction of three resonances 218 involving the glucose, MS and cavity, to yield a potentially richer polaritonic physics beyond the standard 219 upper and lower polariton states typical of strong coupling experiments.” (page 7, lines 218-220).

In Figure 5b they discuss the situation where “the plasmonic and cavity modes are resonant to each other.” Furthermore, in the spectra of Fig 5a there are always 3 resonances, one of which seems to not participate in the coupling, while the single-cavity model (1) would predict only two resonances.

My concern is that in that case the parameter g_{eff} alone introduced by the authors is not sufficient to describe the system alone. As I stated in my previous reply, the system is rather described as 3 coupled oscillators (two photonic modes, a_1 and a_2). The corresponding Hamiltonian is then:

$$H = \omega_1 a_1^\dagger a_1 + \omega_2 a_2^\dagger a_2 + \omega_{\text{glucose}} B^\dagger B + \Omega(a_1^\dagger + a_1)(a_2^\dagger + a_2) + g_1(a_1^\dagger + a_1)(B^\dagger + B) + g_2(a_2^\dagger + a_2)(B^\dagger + B)$$

So, there are actually 3 coupling constants, g_1 , g_2 and Ω that must be evaluated. The constant Ω alone can be evaluated by simulating the transmission maps in a panel like Figure 4 (right) as function of Δd without the resonant contribution of the glucose. Looking at the results from Fig 5b, One can assume that Ω should be very weak. In that case the first three terms of the above Hamiltonian, for $\omega_1 = \omega_2$ can be diagonalized in a symmetric and anti-symmetric resonance, one of which is coupled with the glucose with an enhanced constant $(g_1 + g_2)/\sqrt{2}$, while the other being essentially in a weak coupling and yielding the middle resonance in the spectra of Figure 5a.

Perhaps the authors could provide an analysis in that direction: that will help increase the clarity of the paper and provide a more rigorous ground for their claims, and ultimately increase the impact of their work. That would be my final concern before agreeing with publishing in Nature Communications.

Concerning my remark on electromagnetic simulations and the authors reply: of course, the quantum theory is in accordance with the electromagnetic simulations. My point was that electromagnetic simulations automatically take into account the complexity of the system. A quantum description based on Hamiltonian like (1) requires to identify precisely all the resonant modes and their mutual couplings in order to account for that complexity, in order to have a correct description of the system.

We appreciate the continued effort of the Reviewer in helping us to improve the presentation of our results to the readers. In particular, we found the latest remarks regarding the description of strong coupling from a model going beyond the single-mode Tavis-Cummings Hamiltonian very useful in trying to further clarify the novelty of the hybrid cavity approach. In this revised version of the manuscript, we added this discussion suggested by the Reviewer, which we believe helps clarify this aspect of hybrid transverse-longitudinal modes and their importance in improving the physical description of strong light-matter coupling.

In this document, the Reviewer's comments are in **black**, our responses are in **blue**, sections of the manuscript are in **purple**, and all changes to the manuscript are in **red**.

The main difficulty with the paper is the presentation of the results, which still persist in Figure 4 of the paper. The problem is the complexity of the structure, which presents many different modes that appear either as peaks (FP modes) or dips (MS modes), if I currently understand correctly their description. The light-matter coupling constant, as defined by the authors through the Hamiltonian (1), can be reliably extracted from spectroscopic data only in the case where there is a single mode interacting with the matter excitation, and all other modes of the system are sufficiently far so not to perturb this simple picture (see for instance <https://journals.aps.org/prl/abstract/10.1103/PhysRevLett.105.196402>, or <https://journals.aps.org/prb/abstract/10.1103/PhysRevB.86.125314>).

My main concern, which have not been addressed yet, is the proper estimation of the light-matter coupling constant in the case of the hybrid MS-FP structure. The Hamiltonian (1) corresponds to the case of a single cavity resonance coupled with the glucose transition. In that case, there is a single coupling parameter g_{eff} that can be extracted from the data and compared to other system.

However, in the case of FP-MS resonator the situation seems to be more complex, and the model (1) does not apply any more. The authors acknowledge that fact: they state in the main text “[strong coupling]... Another arises from the interaction of three resonances 218 involving the glucose, MS and cavity, to yield a potentially richer polaritonic physics beyond the standard 219 upper and lower polariton states typical of strong coupling experiments.” (page 7, lines 218-220).

R1: The Referee is perfectly correct that the hybrid architectures that we aim to describe are more complex than a two mode hybridization between light and matter modes. This is of course one of the central messages of our manuscript in that we aim at providing a ‘hybrid approach’ to the electromagnetic degree of freedom, which is constructed from the interference of a cavity-supported longitudinal mode and a transverse, MS-supported mode. In view of this, the Referee is correct that our previous formulation lacked preciseness and we are improving now our presentation (see R2 below for additions to the manuscript). We also agree that the sentence “[...] the interaction of three resonances [...] yield a potentially richer polaritonic physics beyond the standard upper and lower polariton states [...]” can lead to confusion as our results can be accurately described by a single pair of lower and upper polaritons in our experiments. **We have therefore removed this sentence to prevent confusion.**

In Figure 5b they discuss the situation where “the plasmonic and cavity modes are resonant to each other.” Furthermore, in the spectra of Fig 5a there are always 3 resonances, one of which seems to not participate in the coupling, while the single-cavity model (1) would predict only two resonances.

My concern is that in that case the parameter g_{eff} alone introduced by the authors is not sufficient to describe the system alone. As I stated in my previous reply, the system is rather described as 3 coupled oscillators (two photonic modes, a_1 and a_2). The corresponding Hamiltonian is then:

$$H = \omega_1 a_1^\dagger a_1 + \omega_2 a_2^\dagger a_2 + \omega_{\text{glucose}} B^\dagger B + \Omega (a_1^\dagger + a_1) (a_2^\dagger + a_2) + g_1 (a_1^\dagger + a_1) (B^\dagger + B) + g_2 (a_2^\dagger + a_2) (B^\dagger + B)$$

So, there are actually 3 coupling constants, g_1 , g_2 and Ω that must be evaluated. The constant Ω alone can be evaluated by simulating the transmission maps in a panel like Figure 4 (right) as function of Δd without the resonant contribution of the glucose. Looking at the results from Fig 5b, One can assume that Ω should be very weak. In that case the first three terms of the above Hamiltonian, for $\omega_1 = \omega_2$ can be diagonalized in a symmetric and anti-symmetric resonance, one of which is coupled with the glucose with an enhanced constant $(g_1 + g_2)/\sqrt{2}$, while the other being essentially in a weak coupling and yielding the middle resonance in the spectra of Figure 5a.

Perhaps the authors could provide an analysis in that direction: that will help increase the clarity of the paper and provide a more rigorous ground for their claims, and ultimately increase the impact of their work. That would be my final concern before agreeing with publishing in Nature Communications.

R2: We would like to stress the logic of our presentation. We introduce the Tavis-Cummings Hamiltonian in Eq. (1) in order to pedagogically introduce the physics of strong light-matter coupling between a generic single light mode and a collective matter operator. As correctly pointed out by the Referee, this is applicable only to the most simple architectures presented in our manuscript, notably the MS-glucose and planar cavity-glucose interactions. Subsequently, taking into account the Referee's suggestion to improve the theoretical description of hybrid architectures, we introduce in this revised version a Hamiltonian which accounts for the MS and cavity interaction and complements our combined transfer matrix and FDTD analysis of the hybrid architectures. The hybrid architectures are first considered with an empty cavity where the interference of the MS transverse mode and the cavity longitudinal mode leads to the emergence of a modified photonic mode. The theory, classical and quantum for this approach, is presented step-by-step in our theory paper PRL 122, 243601 (2019) (for Fano mirrors) and then in PRX Quantum 3, 010201 (2022) (for more general MSs). The Reviewer will find that the coupling interactions in this configuration are non-trivial and, maybe counterintuitively, do not correspond to a weak coupling strength Ω as suggested by the Reviewer, but rather to a strong one. This aligns with our experiments on the empty hybrid cavity (see Fig. S6 of Suppl Info.), showing a strongly modified transmission due to mode coupling. In our transfer matrix calculations, the fitting parameter g_{eff} must be enhanced for H2 to retrieve the same modified transmission as the one observed experimentally.

The Reviewer makes a good point that our claims could be strengthened by providing information about a more detailed quantum approach to understanding the H2 architecture. To that end, we have elected to include an expression for a hybrid cavity Hamiltonian in our manuscript within the section "*The nature of the hybrid cavity enhancement*", and a discussion of the complexity of using such a quantum approach to fully characterize the system, including a reference to prior work which has compared the quantum approach to a transfer matrix method.

Addition to the manuscript:

Going beyond the approach of using transfer matrix calculations and FDTD simulations, one can also formulate a quantum description of the hybrid cavity enhancement. The intracavity field of a hybrid cavity architecture can be modeled by considering a coupling interaction between the MS and the planar cavity mode. The Hamiltonian for this system is

$$H = \omega_a a^\dagger a + \omega_d d^\dagger d + \lambda (a^\dagger + a)(d^\dagger + d), \quad (4)$$

where a represents the planar cavity mode, d represents the MS mode, and λ is a coupling parameter³⁴. The interaction described by Eq. (4) yields a new hybrid photonic mode, which is characterized by a single peak in transmission. When glucose is introduced into this architecture, one can then use the hybrid photonic mode in the Tavis-Cummings Hamiltonian of Eq. (1) to describe the coupled system. The coupling strength is then partially determined by the spatial overlap between the glucose and the hybrid field distribution.

Where reference 34 is: Cernotik, O., Dantan, A. & Genes, Phys Rev Lett 122, 243601 (2019)

We now also compare more directly the coupling strength of the hybrid cavity H2 with the one corresponding to the MS-glucose system:

We can observe how in the on-resonant case, the response of the H2 architecture shows an enhancement of the MS-glucose interaction, which leads to an effectively larger Rabi splitting of 200 GHz ($g_{\text{eff}} = 100$ GHz) for the MS-glucose polaritons. The exhibited splitting is now about a factor of two larger than the case of coupling only to the MS.

Concerning my remark on electromagnetic simulations and the authors reply: of course, the quantum theory is in accordance with the electromagnetic simulations. My point was that electromagnetic simulations automatically take into account the complexity of the system. A quantum description based on Hamiltonian like (1) requires to identify precisely all the resonant modes and their mutual couplings in order to account for that complexity, in order to have a correct description of the system.

R3: We appreciate the clarification regarding the previous remark about the quantum theory and electromagnetic simulations.

REVIEWERS' COMMENTS

Reviewer #3 (Remarks to the Author):

The authors have clarified the final concerns, there is no objection to publication.